



**Data supporting the North Atlantic Climate System: Integrated Studies (ACSIS) programme,**

**including** atmospheric composition, oceanographic and sea ice observations (2016-2022) and output

from ocean, atmosphere, land and sea-ice models (1950-2050)

Alex T. Archibald[1, 2], Bablu Sinha[3], Maria R. Russo[1, 2], Emily Matthews[5], Freya A. Squires[6], N. Luke

Abraham[1, 2], Stephane J.-B Bauguitte[4], Thomas. J. Bannan[5], Thomas G. Bell[7], David Berry[8], Lucy J.

Carpenter[9], Hugh Coe[5, 10], Andrew Coward[3], Peter Edwards[9,11], Daniel Feltham[12], Dwayne Heard[13], Jim

Hopkins[9,11], James Keeble[1, 2], Elizabeth C. Kent[3], Brian A. King[3], Isobel R. Lawrence[14,15], James Lee[9,15],

Claire R. Macintosh[16], Alex Megann[3], Bengamin I. Moat[3], Katie Read[9,11], Chris Reed[4], Malcolm J.

Roberts[17], Reinhard Schiemann[18], David Schroeder[12], Timothy J. Smyth[7], Loren Temple[9], Navaneeth

Thamban[5], Lisa Whalley[13,19], Simon Williams[3], Huihui Wu[5], Ming-Xi Yang[7]

[1]National Centre for Atmospheric Science, University of Cambridge, Cambridge, United Kingdom

[2]Yusuf Hamied Department of Chemistry, University of Cambridge, Cambridge CB2 1EW, United Kingdom

[3]National oceanography Centre, United Kingdom.

[4]Facility for Airborne Atmospheric Measurements Airborne Laboratory, Cranfield University, Cranfield MK43 0AL, United

Kingdom

[5]Department of Earth and Environmental Science, Centre for Atmospheric Science, University of Manchester, Manchester

M13 9PL, United Kingdom

[6]British Antarctic Survey, Cambridge CB3 0ET, United Kingdom

[7]Plymouth Marine Laboratory, Plymouth PL1 3DH, United Kingdom

[8]WMO, Geneva, Switzerland

[9]Wolfson Atmospheric Chemistry Laboratories, Department of Chemistry, University of York, York YO10 5DD, United

Kingdom

[10]National Centre for Atmospheric Science, University of Manchester, Manchester M13 9PL, United Kingdom

[11]National Centre for Atmospheric Science, University of York, York, United Kingdom

[12]CPOM, University of Reading, Reading, UK

[13]School of Chemistry, University of Leeds, Leeds LS2 9JT , United Kingdom

[14]ESA ESRIN, Via Galileo Galilei, 1, 00044 Frascati RM, Italy.

[15]CPOM, University of Leeds, Leeds, United Kingdom

[16]ESA Climate Office, United Kingdom

[17]Met Office Hadley Centre, Exeter, UK.

[18]National Centre for Atmospheric Science, Department of Meteorology, University of Reading, Reading, UK

[19]National Centre for Atmospheric Science, University of Leeds, Leeds, United Kingdom



*Correspondence to*: Alex Archibald (ata27@cam.ac.uk) and Bablu Sinha (bablu@noc.ac.uk)

**Abstract.** The North Atlantic Climate System: Integrated Study (ACSIS) was a large multidisciplinary research programme

funded by the United Kingdom's Natural Environment Research Council (NERC). ACSIS ran from 2016-22 and brought

together around 80 scientists from seven leading UK-based environmental research institutes to deliver major advances in

understanding North Atlantic climate variability and extremes. Here we present an overview of the data generated by the

ACSIS programme. The datasets cover the full North Atlantic System comprising: the North Atlantic Ocean, the atmosphere

above it including its composition, Arctic Sea Ice and the Greenland Ice Sheet.

Atmospheric composition datasets include measurements from 7 aircraft campaigns (between 3 and 10 flights each, 0-10km

altitude range) in the north eastern Atlantic (~40°W-5°E,~15°N-55°N) made at intervals of from 6 months to 2 years between

February 2017 and may 2022. The flights measured chemical species (including greenhouse gases, ozone precursors and

VOCs) and aerosols (organic, $SO_4$, $NH_4$, $NO_3$, and nss-Cl) (https://dx.doi.org/10.5285/6285564c34a246fc9ba5ce053d85e5e7

(FAAM et al. (2024)). Ground based stations at the Cape Verde Atmospheric Observatory (CVAO), Penlee Point Atmospheric

Observatory (PPAO) and Plymouth Marine Laboratory (PML) recorded ozone, ozone precursors, halocarbons, as well as

greenhouse          gases          ($CO_2$,          methane),          $SO_2$          and          photolysis          rates.          (CVAO,

http://catalogue.ceda.ac.uk/uuid/81693aad69409100b1b9a247b9ae75d5, National Centre for Atmospheric Science et al.

(2014)), $O_3$ and $CH_4$ (PPAO, https://catalogue.ceda.ac.uk/uuid/8f1ff8ea77534e08b03983685990a9b0 (Plymouth Marine

Laboratory and Yang (2024)) and aerosols (PML, https://dx.doi.org/10.5285/e74491c96ef24df29a9342a3d57b5939, Smyth

(2024)).

Complementary model simulations of atmospheric composition were performed with the UK Earth System Model, UKESM1,

for the period 1982 to 2020 using CMIP6 historical forcing up to 2014 and SSP3-7.0 scenario from 2015-2020. Model

temperature and winds were relaxed towards ERA reanalysis. Monthly mean model data for ozone, NO, $NO_2$, CO, methane,

stratospheric          ozone          tracers          and          30          regionally          emitted          tracers          are          available          to          download

(https://data.ceda.ac.uk/badc/acsis/UKESM1-hindcasts, Abraham (2024)).

ACSIS also generated new ocean heat content diagnostics https://doi.org/10/g6wm, https://doi.org/10/g8g2, Moat et al.

(2021a-b)          and          gridded          temperature          and          salinity          based          on          objectively          mapped          Argo          measurements

https://doi.org/10.5285/fe8e524d-7f04-41f3-e053-6c86abc04d51 (King (2023).

An ensemble of atmosphere-forced global ocean-sea ice simulations using the NEMO-CICE model was performed with

horizontal resolutions of ¼° and 1/12° covering the period 1958-2020 using several different atmosphere reanalysis based



surface forcing datasets, supplemented by additional global simulations and standalone sea ice model simulations with
advanced sea ice physics using the CICE model (http://catalogue.ceda.ac.uk/uuid/770a885a8bc34d51ad71e87ef346d6a8,
Megann et al. (2021e). Output is stored as monthly averages and includes 3D potential temperature, salinity, zonal, meridional
and vertical velocity; 2D sea surface height, mixed layer depth, surface heat and freshwater fluxes, ice concentration and
thickness and a wide variety of other variables.
In addition to the data presented here we provide a brief overview of several other datasets that were generated during ACSIS
which have been described previously in the literature.

## 1. The North Atlantic Climate System

The North Atlantic Climate System Integrated Study (ACSIS) was a 6 year research programme (2016-2022) commissioned
by The UK Natural Environment Research Council (NERC) as part of the first wave of a new series of Long Term Science
Multi-centre (LTSM) projects. ACSIS connected research in the physical and chemical components of the atmosphere-
hydrosphere-cryosphere nexus within the North Atlantic region and provided an opportunity for NERC scientists from
different disciplines to come together and deliver new insights into a region undergoing rapid change in: the ocean and
atmosphere temperatures and circulation, in sea ice thickness and extent, and in key atmospheric constituents such as ozone,
methane and aerosols (Sutton et al., 2018). The ACSIS team included members of the National Centre for Atmospheric Science
(NCAS), Plymouth Marine Laboratory (PML),  the National Oceanography Centre (NOC), the British Antarctic Survey (BAS),
the National Centre for Earth Observation (NCEO), the Centre for Polar Observation and Modelling (CPOM), and the Met
Office.
ACSIS was designed to answer key questions about the North Atlantic Climate System:
1) How have changes in natural and anthropogenic emissions and atmospheric circulation combined to shape multi-year trends
in North Atlantic atmospheric composition and radiative forcing? 2) How have natural variability and radiative forcing
combined to shape multi-year trends in the North Atlantic physical climate system? 3) To what extent are changes in the North
Atlantic climate system predictable on multi-year timescales?
In order to answer these questions, ACSIS was arranged into a series of interlinked work packages involving a broad
representation of scientists from the different NERC centres involved in ACSIS. These work packages delivered new scientific
understanding, delivered through several key synthesis papers (Sutton et al., 2018, Robson et al., 2018, 2020, Hirschi et al.,
00  2020)  as well as a wealth of data. Key objectives of the ACSIS project were to:
01  A) Provide the UK science community with sustained observations, data syntheses, leading-edge numerical simulations and
02  analysis tools to facilitate world-class research on changes in the North Atlantic climate system and their impacts. B) To





03 provide a quantitative and multivariate description of how the North Atlantic climate system is changing. C) To determine the

04 primary drivers and processes that are shaping changes in the North Atlantic climate system now and will shape changes in

05 the near future. D) To determine the extent to which future changes in the North Atlantic climate system are predictable.

06 In this paper we focus on objective (A) of the ACSIS project, which included the creation of new datasets to underpin the

07 ACSIS project and support wider work on the North Atlantic climate system by the UK and international science communities.

08

09 In this paper we outline the underpinning datasets generated as part of the ACSIS project, how they can be obtained (guided

10 by the FAIR principles (Wilkinson et al., 2016)), and the motivation for their creation.

11 **1.1 Overview of data holdings**

12 A summary of the datasets that are generated by ACSIS and freely available to the community is given in Table 1. Note that

13 the new data presented in this paper are archived across two platforms: the British Oceanographic Data Centre,

14 https://www.bodc.ac.uk (ocean observations) and the Centre for Environmental Data Analysis, https://www.ceda.ac.uk (all

15 other data) . A schematic map giving an overview of the footprints of all the observational datasets can be found in Fig 4 of

16 Sutton et al. (2018). The three general areas covered are: atmospheric composition covering aircraft and ground station data

17 along with nudged historical atmospheric chemistry/circulation model simulations; ocean observations covering gridded in

18 situ temperature and salinity (0-2000m) and 0-1000m heat content;  forced historical ocean-ice simulations at eddy permitting

19 and eddy resolving resolutions and standalone Arctic sea ice simulations. In subsequent sections 2, 3 and 4,  we describe the

20 individual archived datasets in detail. Several other datasets, previously described in the literature,  have been generated by the

21 ACSIS programme including simulations to generate volcanic forcing data for climate models, coupled climate model

22 simulations with a high resolution atmosphere and/or ocean, gridded sea-surface temperature based on in situ ocean

23 observations, and observation based estimates of the Atlantic Meridional Overturning Circulation and Arctic wide sea ice

24 thickness. For completeness, and because the new datasets described here will likely be used in conjunction with the already

25 published datasets, we provide a brief overview of the latter in Section 5.

27 **Table 1**. Overview of the data described in this paper with links to the sub-sections where the data are described in detail.

| Title | Data, weblink, and citation | Accessibility | Subsection |
|---|---|---|---|
| Aircraft missions | Gas and aerosol data collected on board the Facility for Airborne Atmospheric Measurements https://dx.doi.org/10.5285/6285564c34a246fc9ba5ce053d85e5e7 FAAM et al. (2024) | Open access for merged 10s data; registration/login to CEDA | 2.1 |



| | | required for full temporal resolution. | |
|---|---|---|---|
| Ground based observational atmospheric composition time series | Atmospheric composition, including ozone, methane, carbon monoxide, VOCs and aerosol parameters from the Cape Verde Atmospheric Observatory (CVAO) http://catalogue.ceda.ac.uk/uuid/81693aad69409100b1b 9a247b9ae75d5 National Centre for Atmospheric Science et al. (2014) Penlee Point Atmospheric Observatory (PPAO) https://catalogue.ceda.ac.uk/uuid/8f1ff8ea77534e08b03 983685990a9b0 Plymouth Marine Laboratory and Yang (2024). Plymouth Marine Laboratory https://catalogue.ceda.ac.uk/uuid/e74491c96ef24df29a9 342a3d57b5939 Smyth (2024) | CVAO data require registration/login to CEDA. PPAO and PML data are open access. | 2.2, 2.3 |
| Nudged atmosphere model simulations with atmospheric composition | Simulated atmospheric composition from 1981-2020 with atmospheric circulation nudged to ERA5 reanalysis https://data.ceda.ac.uk/badc/acsis/UKESM1-hindcasts Abraham (2024) | Open access for selected atmospheric composition variables. Requires registration/login on JASMIN and Met Office MASS account for access to comprehensive dataset. | 2.4 |
| Ocean circulation and heat content | Objectively interpolated (gridded) ocean temperature and salinity (0-2000m) https://doi.org/10.5285/fe8e524d-7f04-41f3-e053-6c86abc04d51 King (2023) Upper Ocean (0-1000m) heat content time series | Open access. | 3.1, |



| Ocean-sea ice and standalone sea ice simulations | NEMO-CICE global ocean simulations with default sea ice physics $1^{o}$, $1/4^{o}$ and $1/12^{o}$ up to 2020 https://dx.doi.org/10.5285/119a5d4795c94d2e94f610647640edc0  Megann et al. (2021b) https://dx.doi.org/10.5285/a0708d25b4fc44c5ab1b06e12fef2f2e,  Megann et al (2021c) https://dx.doi.org/10.5285/4c545155dfd145a1b02a5d0e577ae37d,  Megann et al. (2021d) https://dx.doi.org/10.5285/e02c8424657846468c1ff3a5acd0b1ab Megann et al. (2022a) https://dx.doi.org/10.5285/399b0f762a004657a411a9ea7203493a (Megann et al. (2022b)  NEMO-CICE global ocean simulations with improved sea ice physics $1/4^{o}$ up to 2020 and standalone Arctic sea ice simulations:  http://catalogue.ceda.ac.uk/uuid/770a885a8bc34d51ad71e87ef346d6a8 Megann et al. (2021e) | open access | 3.2.2, 4.1 |

## 2. Composition data sets

The composition of the atmosphere is changing at an unprecedented pace. Changes in the levels of stratospheric ozone, surface ozone and other secondary pollutants are driven by human activities (e.g., Griffiths et al., 2021; Keeble et al., 2020; Turnock et al., 2020). The North Atlantic region has undergone significant growth and decline in air pollution over the last three decades and modelling studies have all shown the significant human health benefits of these more recent reductions (Turnock et al. 2016; Archibald et al., 2017; Daskalakis et al., 2016). But whilst we have a broad understanding of the distribution of key air pollutants and short lived climate forcers, our understanding of the variability of these species and their trends is hampered across the North Atlantic owing to a paucity of observations. The North Atlantic is frequently impacted by the transport of



transboundary pollution from anthropogenic sources and fires (Boylan et al., 2015; Helmig et al., 2015; Kumar et al., 2013),

as well as from local natural marine and shipping emissions (e.g., Yang et al., 2016a). High altitude research stations in the

Eastern North Atlantic in the Azores (Mt. Pico) and Canary Islands (Izána), coastal observatories on the west coast of Ireland

(Mace Head) and in the Cape Verde Islands have provided long term data sets with which to better understand the sources and

processes controlling reactive trace gases and aerosols across the North Atlantic.

In ACSIS a series of work packages were conducted to a) further our understanding of the distribution and variability of key

trace gases and aerosols using aircraft campaigns and long-term measurements, b) understand the processes controlling these

and c) improve model simulations, which can be used to forecast the future evolution of these species. In the following sections

we outline the data that were generated to support these objectives.

### 2.1 Aircraft campaigns in the North Atlantic

A series of (daytime) research flights were carried out across the North Atlantic Ocean from February 2017 – May 2022.

Measurements were collected using the UK's Atmospheric Research Aircraft (ARA). The ARA is a BAe-146-301 which has

been in service since 2004 and is managed by the Facility for Airborne Atmospheric Measurements (FAAM),  an airborne

laboratory funded by the UK government. The FAAM aircraft is capable of carrying a 4-tonne instrument load and can operate

at altitudes between 50 and 30000 ft (15–9140 m), allowing the study of processes in the troposphere and boundary layer.

ARA missions as part of ACSIS provide the longest record of composition change in the lower free troposphere over the North

Atlantic (Sutton et al., 2018) and further complemented historic research flights conducted with the ARA in the region (e.g.,

Parrington et al., 2012; Reeves et al., 2002) and more recent flights by other platforms (e.g., ATom (Wofsy et al., 2018),

NAAMES (e.g., Behrenfeld et al., 2019; Sinclair et al., 2020) and ACE-ENA (Zawadowicz et al., 2021)). A wide range of

instrumentation are fitted on the ARA, including measurements of key meteorological parameters such as temperature,

humidity, wind speed and direction as well as a range of in situ trace gas measurements including carbon monoxide (CO),

ozone ($O_3$), oxides of nitrogen ($NO_x=NO+NO_2$), and the greenhouse gases carbon dioxide ($CO_2$) and methane ($CH_4$). Table 2

below summarises the measurement techniques and uncertainties onboard the ARA that were used during ACSIS flights.

**Table 2.** A summary of atmospheric chemistry instrumentation used during the ACSIS flights onboard the FAAM BAe-146-301 Atmospheric Research Aircraft.

| Measurement | Instrumentation | Time resolution | Precision 3σ | Uncertainty | Timescale | Data available in merge file |
|---|---|---|---|---|---|---|
| $O_3$ | Thermo 49i ozone photometer | 4 sec | 6 ppb | 3 ppb / 3% | 2017-2021 | X |
| $O_3$ | 2BTechnologies Model 205 ozone photometer | 2 sec | 4 nmol mol$^{-1}$ | 5 ppb / 3% for $O_3$ > 100 nmol mol$^{-1}$ | 2022-present | X |





| | | | | | | |
|---|---|---|---|---|---|---|
| CO | AeroLaser AL5002 (VUV RF) | 1 sec | 6 ppb | 2 ppb | 2005-2019 | X |
| $CO_2$ | Los Gatos Research FGGA (OA-ICOS) | 1 sec | 1.5 ppm | 0.5 ppm | 2011-present | X |
| $CH_4$ | Los Gatos Research FGGA (OA-ICOS) | 1 sec | 6 ppb | 3 ppb | 2011-present | X |
| NO | Chemiluminescence Air Quality Design Inc | 10 sec | 10 ppt | 24% | 2009-2019 | X |
| $NO_2$ | Chemiluminescence Air Quality Design Inc | 10 sec | 13 ppt | 41% | 2009-2019 | X |
| NO | Chemiluminescence Air Quality Design Inc (upgraded) | 0.1 sec | 30 ppt | 24% | 2019-present | X |
| $NO_2$ | Chemiluminescence Air Quality Design Inc (upgraded) | 0.1 sec | 60 ppt | 41% | 2019-present | X |
| $SO_2$ | University of York laser-induced fluorescence sulfur dioxide detector (LIF-SO2) | 1 sec | 225 ppt | 15 % | 2022-present | X |
| Solar Actinic flux | Ocean Optics QE Pro, up and downward facing UV-vis (280-700 nm) spectrometers | 1 sec | TBC | 5 % | 2019-present | X |
| HCHO | LIF pulsed 353.370 nm spectrometer, Thermo Scientific Model TFL 3000 Novawave | 1 sec | n/a | n/a | 2019-present | |
| VOCs | Whole Air Samples and offline analysis by GC-FID or GC-MS | n/a | | | 2005-present | |
| Other gases | University of Manchester High Resolution-Time of Flight-Chemical Ionisation Mass Spectrometer (ToF-CIMS) | 0.25 sec | | 10-20% | 2019-present | |
| HONO | ToF-CIMS | 0.25 sec | n/a | 20% | | |
| HCN | ToF-CIMS | 0.25 sec | | 30% | | X |
| BrO | ToF-CIMS | 0.25 sec | n/a | 40% | | |
| BrCl | ToF-CIMS | 0.25 sec | n/a | 40% | | |
| $ClNO_2$ | ToF-CIMS | 0.25 sec | | 30% | | X |
| $Cl_2$ | ToF-CIMS | 0.25 sec | n/a | 20% | | |
| ClO | ToF-CIMS | 0.25 sec | n/a | 40% | | |
| HPMTF § | ToF-CIMS | 0.25 sec | n/a | n/a | | |
| Urea | ToF-CIMS | 0.25 sec | 30 ppt | 25% | | X |



| Aerosol | University of Manchester Aerosol Mass Spectrometer (AMS) | | | | 2019-present (excl. 2020) | |
|---|---|---|---|---|---|---|
| Organic | AMS | 8-15 sec | 0.03 µg/m³ | 38% | | X |
| SO4 | AMS | 8-15 sec | 0.03 µg/m³ | 36% | | X |
| NH4 | AMS | 8-15 sec | 0.03 µg/m³ | 34% | | X |
| NO3 | AMS | 8-15 sec | 0.03 µg/m³ | 34% | | X |
| nss-Cl | AMS | 8-15 sec | 0.03 µg/m³ | n/a | | X |

§Hydroperoxy methyl thioformate.

Figure 1 shows the location of the ACSIS flight tracks, coloured by campaign number. There were a total of 45 flights as part

of the ACSIS campaign, comprising close to 200 hours of measurement data. Measurements were made from approximately

50 m over the sea surface to 7600 m. ACSIS 1, 2, 4, 5 and 7 were predominantly based out of the Azores, whilst flights for

ACSIS 3 were based out of Cork, Ireland and ACSIS 6 flights based out of Cape Verde.



**Figure 1**. A map of flight tracks for the seven ACSIS ARA campaigns. Part of the NASA ATom flight campaign flight tracks
are shown in grey for comparison.





Also shown in Fig. 1 are part of the flight tracks for the NASA Atmospheric Tomography Mission (ATom) mission. The
ATom campaigns aimed to improve the representation of reactive gases and short-lived climate forcers in global atmospheric
chemistry and climate models by measuring atmospheric composition along a global circuit flight track (Prather et al., 2017).
Four ATom campaigns occurred between August 2016 and May 2018. The ATom data set is complementary to that collected
during the ACSIS flight campaigns; ATom flights provided a broad overview on a global scale, whereas ACSIS flights
intensively measured the North Atlantic region. ACSIS-1 overlapped with ATom2 and ACSIS-2 overlapped with ATom3.

*2.1.1 Bulk Analysis of Data*

Data collected during flights from all 7 ACSIS campaigns have been analysed together to give insights into the spatial and
vertical characteristics of atmospheric composition over the North Atlantic Ocean. Table 3 summarises the flights and times
that were used in this bulk analysis.

**Table 3.** Summary of flights used in bulk analysis of atmospheric composition data.

| Campaign | Flight Numbers | Date Range | Comments |
|---|---|---|---|
| ACSIS 1 | B996, B997, B998, B999, C001, C002 | 13/02/2017 – 16/02/2017 | |
| ACSIS 2 | C066, C067, C068, C070, C071 | 19/10/2017 – 23/10/2017 | |
| ACSIS 3 | C103, C105, C106 | 14/05/2018 – 17/05/2018 | No greenhouse gas data available due to the FGGA fault. No VOC data on CEDA |
| ACSIS 4 | C139, C140, C141, C142, C143, C144, C145 | 19/02/2019 – 22/02/2019 | |
| ACSIS 5 | C199, C200, C201, C202, C203, C204, C205, C210, C211, C212 | 13/08/2019 – 22/08/2019 | |
| ACSIS 6 | C215, C216, C217, C226, C227, C228, C229 | 04/02/2020 – 14/02/2020 | . |
| ACSIS 7 | C288, C289, C290, C291, C292, C293, C294 | 03/05/2022 – 09/05/2022 | |

88

89  *2.1.3 Vertical Distribution of Pollutants*

90  Data from all seven campaigns have been combined and grouped into 1000 m altitude bins. Figure 2 shows the vertical
91  distribution of $O_3$, CO, $CO_2$, $CH_4$, NO and $NO_2$.







**Figure 2.** Box plots showing the vertical distribution of $O_3$, CO, $CO_2$, $CH_4$, NO and $NO_2$ for all seven ACSIS campaigns. Note that due to sporadic high mixing ratios of CO, NO and $NO_2$ at low altitudes these data have been filtered to only show data below 600 ppbv for CO and  500 pptv for NO and $NO_2$.



### 2.1.2 Data archive

To accompany this paper a 10 second averaged merge file has been created for each flight listed in Table 3 (https://dx.doi.org/10.5285/6285564c34a246fc9ba5ce053d85e5e7, Facility for Airborne Atmospheric Measurements et al., 2024). The merge files are open access and designed to be a tool for an initial exploration of the data and to highlight the breadth of the atmospheric composition data collected during the ACSIS programme. However, for further analysis the original frequency data should be used and details of where these files can be found is included in the header information of the merge files The merged files are in ascii format and consist of a short explanatory paragraph followed by a list of variables and finally the data arranged as columns, with one variable per column with rows corresponding to the values at each 10s time interval. .

## 2.2 Cape Verde Atmospheric Observatory (CVAO)

The Global GAW Cape Verde Atmospheric Observatory is situated in Calhau on the island of Sao Vicente in the Republic of Cabo Verde (16.848°N, 24.871°W, 10m asl, https://amof.ac.uk/observatory/cape-verde-atmospheric-observatory-cvao/). Measurements were started in October 2006 to further our understanding of atmospheric chemistry within the tropical marine boundary layer and North Atlantic region. The site receives air from a wide variety of sources with 10-day back trajectories reaching to North America, Europe and sub-Saharan Africa (see Carpenter et al. (2010) for details). Long term high frequency measurements allow investigation into the trends of climate gases such as $CO_2$ and $CH_4$ whilst measurements of pollutants from the continents such as hydrocarbons and nitrogen oxides provide better constraints of global emission changes and their effect on the long-term background of the North Atlantic (e.g., Helmig et al., 2016). The Observatory regularly hosts field campaigns which focus on process studies such as sea-surface interactions and the role of aerosols in atmospheric chemistry (Read et al., 2008, McFiggans et al., 2009, Lawler et al., 2011, Van Pinxteren et al., 2020). Recent work has provided evidence for the rapid photolysis of nitrate aerosol as an important source of HONO and $NO_2$ to the marine troposphere (Andersen et al., 2021). Andersen et al. (2023) confirmed the ubiquity of this so-called "renoxification" source of HONO and NOx in the remote Atlantic troposphere using HONO observations at the CVAO and from aircraft in the surrounding remote Atlantic troposphere and showed evidence for renoxification occurring on marine and mixed marine/dust and biomass burning aerosols with an efficiency that increased with relative humidity and decreased with the concentration of nitrate. Table 4 provides a summary of the chemical species recorded at the CVAO.

**Table 4.** Summary of atmospheric data recorded at CVAO.

| Measurement | Instrumentation | Time resolution | Precision (1hr) | Timescale |
|---|---|---|---|---|
| $O_3$ | Thermo 49i ozone monitor | 10 sec | 0.5 ppb | 2006-present |
| CO | Aerolaser AL5001/ Picarro G4201 | 4 sec | 1 ppb | 2008-present |



| | | | | |
|---|---|---|---|---|
| NO | Chemiluminescence instrument Air Quality Design Inc. (AQD), USA | 5 min | 1.4 ppt | 2006-present |
| NO$_2$ | Chemiluminescence instrument Air Quality Design Inc. (AQD), USA | 5 min | 4.4 ppt | 2017-present |
| VOCs | GC-FID | 1 hour | | 2006-present |
| OVOCs | GC-FID | 1 hour | | 2014-present |
| Short-lived halocarbons | GC-MS-TOF | 1 hour | | 2014-present |
| CFCs/HCFCs | GC-MS-TOF | 1 hour | | 2022-present |
| DMS | GC-FID | 1 hour | | 2012-present |
| Photolysis rates | Spectral radiometer | 1 min | | 2016-present |
| CO$_2$ | Picarro G4201 | 4 sec | 10 ppb | 2012-present |
| CH$_4$ | Picarro G4201 | 4 sec | 0.3 ppb | 2012-present |
| SO$_2$ | Thermo 43i HL | 5 sec | | 2019-present |
| Total Gaseous Mercury | Tekran | 1 min | | 2014-2019 |

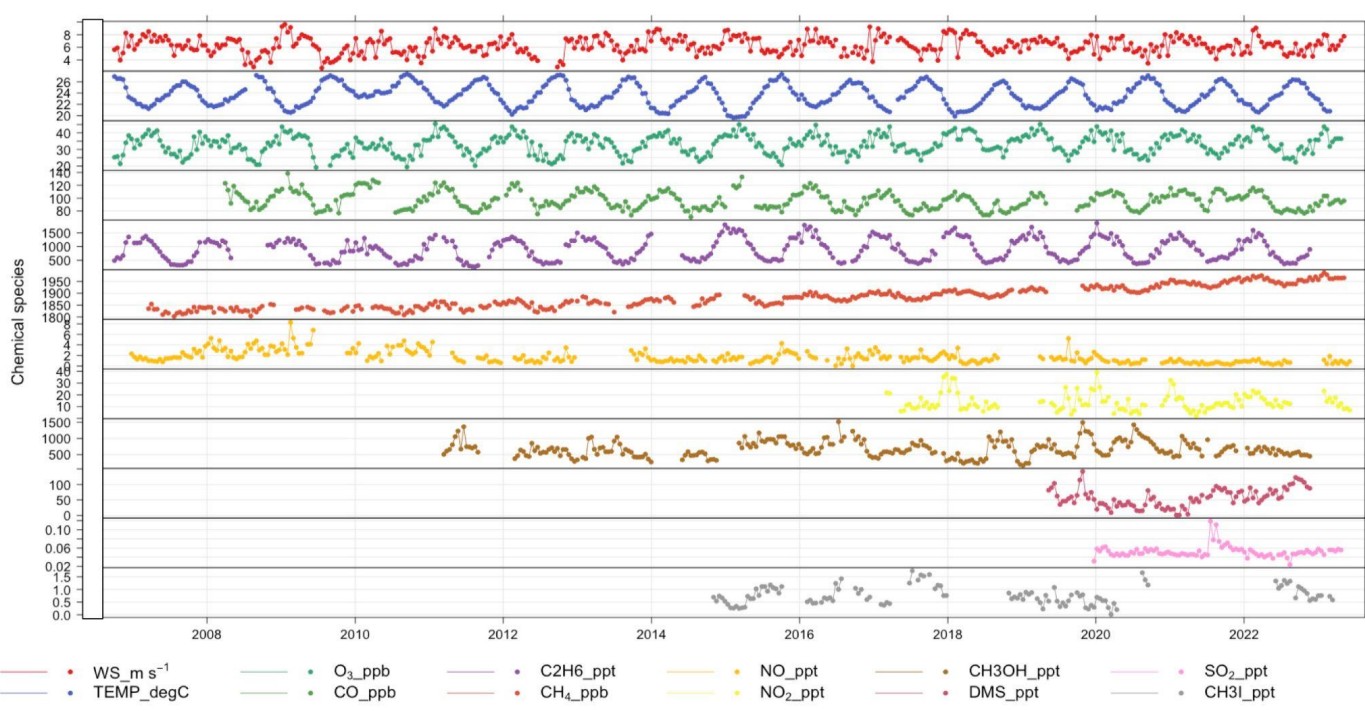





**Figure 3.** Time series of Cape Verde data showing a range of the species and meteorological parameters measured..

### 2.2.1 Ozone

Ozone concentrations at the CVAO show seasonal variability with highest concentrations in spring and lowest in summer, consistent with its role as a secondary pollutant. During the period 2007-2022, ozone steadily increased at a rate of 0.16 +/- 0.08 ppbV/year. In summer, the site occasionally receives air from the southern hemisphere during the early stages of the Atlantic cyclonic activity, which leads to very low concentrations of ozone (<10 ppb) observed along with episodes of intense precipitation. Ozone data from the CVAO contributed to the Tropospheric Ozone Assessment Report (TOAR) (Schultz et al., 2017).

### 2.2.2 Carbon monoxide

Carbon monoxide is a primary pollutant emitted from anthropogenic sources and from biomass burning. Between 2008-2016 levels were decreasing at a rate of 0.6 +/- 0.8 ppbV/ year. Since 2016 CO has been decreasing at a slower rate of 0.14 +/- 0.9 ppbV/year.

### 2.2.3 Methane

Global methane concentrations have increased substantially over the last 10 years and this has been attributed to a combination of increased primary emissions of hydrocarbons and increased emissions from wetlands due to increasing temperatures (Jackson et al, 2020, Thompson et al., 2018). At the CVAO methane has been increasing steadily (8 +/- 0.4 ppbV/year) however in the last couple of years (2020-2023) the rate has increased by almost 60% to 14 +/- 1.8 ppbV/year. Observations of ethane and other hydrocarbons can give an indication as to whether changes in methane are due to changes in emissions or in oxidant levels and whether increased emissions are natural or anthropogenic (Helmig et al, 2016). Ethane shows an increasing trend of 17.5 +/- 5 pptV until 2020 and then strongly decreases after that 27.1 +/- 17 pptV.

.

### 2.2.4 $NO_x$

NOx mixing ratios at the CVAO peak around solar noon at 20–30 pptv, in contrast to model simulations that predict a midday minimum due to conversion of $NO_2$ to $HNO_3$ through reaction of OH (Reed et al. 2017). Inclusion of the rapid photolysis of nitrate aerosol to produce HONO and $NO_2$ is required to successfully simulate the observed diurnal cycles of these gases at the CVAO (Reed et al. 2017; Andersen et al. 2023). In extremely clean air containing low levels of CO and VOCs, Andersen et al. (ACP, 2022) showed good agreement between NO2 levels observed at the CVAO and those derived from the photostationary state (PSS), utilising measured NO, O3, and jNO2 and photo-chemical box model predictions of peroxy radicals. However, in clean air containing small amounts of aged pollution, as typically encountered in winter, higher levels



of NO2 were observed than inferred from the PSS, implying underestimation of peroxy radicals or unattributed NO2

measurement artefacts.

### 2.2.5 Data archive

Cape Verde data collected under the auspices of ACSIS is available from CEDA:

http://catalogue.ceda.ac.uk/uuid/81693aad69409100b1b9a247b9ae75d5 (National Centre for Atmospheric Science et al.

(2014)). Note that there are a number of subdirectories, some of which are not relevant to the data described in this paper. The

relevant subdirectories are labelled with the variable or variable group and the time period (e.g. Cape Verde Atmospheric

Observatory: Ozone measurements (2006 onwards)). The data format is ASCII, consisting of a header explaining the variables

listed followed by the data in columnar format (one column per variable), with the data values in rows appearing in

chronological order. We note that specific Cape Verde data is also archived at the World Data Centre for Greenhouse Gases,

https://gaw.kishou.go.jp ( $CO_2$, $CH_4$ and CO) and at EBAS, https://ebas.nilu.no (VOCs, $NO_x$, $SO_2$ and halocarbons).

### 2.3 Penlee Point Atmospheric Observatory

Penlee Point Atmospheric Observatory (PPAO; 50° 19.08' N, 4° 11.35' W;

https://www.westernchannelobservatory.org.uk/penlee/) was established by the Plymouth Marine Laboratory (PML) in 2014

on the southwest coast of the United Kingdom.  At the western mouth of the Plymouth Sound and near the tip of the Rame

Peninsula, PPAO is a few tens of metres away from the water edge and about 11 m above mean sea level.  The site is exposed

to marine air over a very wide sector.  Typical south-westerly winds tend to bring relatively clean background air coming off

the North Atlantic.  Winds from the southeast are often contaminated by exhaust plumes from passing ships, while winds from

the north are influenced by terrestrial emissions.

In close proximity to the Western Channel Observatory marine sampling stations, high frequency observations at PPAO enable

both long-term and process-based studies of atmosphere-ocean interactions. Current/recent work has assessed trace gas

burdens and air-sea fluxes including greenhouse gases (Yang et al. 2016b, 2016c, 2019a), volatile organic carbon (Phillips et

al., 2021), sulfur- (Yang et al., 2016c), halogen- (Sommariva et al., 2018), and nitrogen-containing gases (ongoing). Further

works include aerosol composition and fluxes, with particular foci on ship emissions (ongoing as a part of the ACRUISE

project), sea spray production (Yang et al., 2019b), macro/micro nutrient deposition (White et al., 2021), and reaction between

atmospheric ozone and the sea surface microlayer (Loades et al., 2020)

Continuous observations most relevant to ACSIS include ground-based ozone and methane from PPAO as well as column

aerosols from the rooftop of PML (10 km north/northeast of PPAO).  These measurements are detailed in Table 5.

**Table 5.** Overview of the measurements made at PPAO.





| Measurement | Instrumentation | Time resolution | Accuracy | Timescale |
|---|---|---|---|---|
| $O_3$ | (a) 2B 205 ozone monitor; (b) Thermo 49i ozone monitor | 10 sec | ≤1 ppb | (a) May 2014 – Sept 2018 (b) Sept 2018 – present |
| $CH_4$ | (a) Picarro G2311-f; (b) Los Gatos Research Fast Greenhouse Gas Analyzer | 0.1 sec until Aug 2016; 1 sec since Aug 2016 | ≤ 3 ppb | (a) May 2014 – Sept 2015 (b) Sept 2015 - present |
| Aerosols | POM sunphotometer | 10 min (when clear sky and during the day) | ≤0.01 at 550 nm | 2001 – present |

### 2.3.1 Ozone

We first examine the ground-based $O_3$ observation, focusing especially on variability within the open ocean (e.g. Atlantic) wind sector. Here we limit our analysis to the first two years of observations (May 2014 to Apr 2016), when airmass dispersion modelling (NAME) was available to assess the airmass history (see Yang and Fleming, 2019). Due to its short lifetime, we expect $O_3$ to be more sensitive than $CH_4$ to local sources/sinks and heterogeneities associated with a coastal environment. This presents a good opportunity to compare two different methods of identifying the open ocean wind sector: 1) by airmass history, and 2) by local wind direction.

Monthly mean $O_3$ observations from the North Atlantic were similar between Penlee Point and Mace Head, both showing spring maxima and summer minima (Figure 4). Defining the PPAO open ocean sector either by local wind direction (210 to 260°) or by airmass history (>80% in the Atlantic Ocean region over the last 5 days) results in fairly consistent monthly mean $O_3$ over most months. The difference between the two methods appears to be slightly larger in summer; possibly due to the diurnal sea breeze effect. In subsequent analyses, we define the open ocean (Atlantic) wind sector by local wind direction only as the NAME modelling for PPAO is unavailable after Mar 2017.





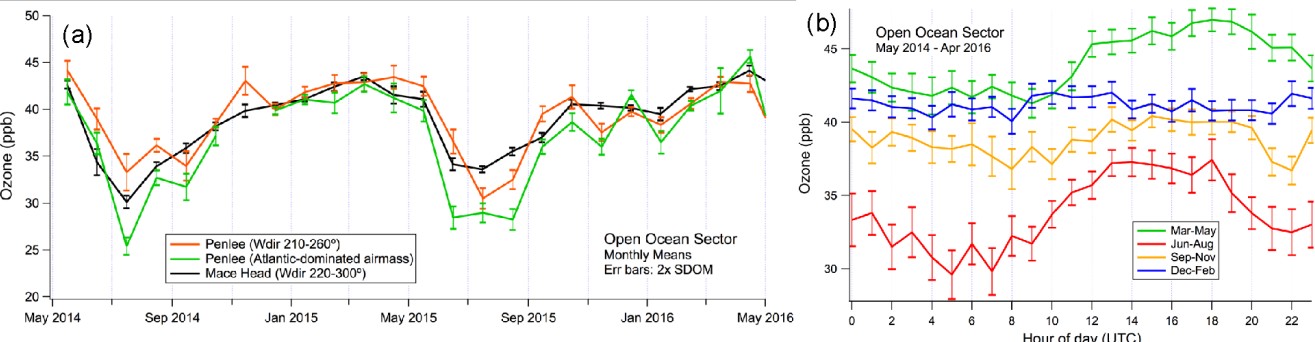

**Figure 4**. (a) Monthly mean ozone mixing ratio from Mace Head and from PPAO (two different definitions of open ocean sector are shown). (b) Diurnal variability in ozone mixing ratio for different seasons. Error bars indicate standard error.

Ozone from the open ocean wind sector shows quite a strong seasonal variability. This is likely due to a combination of variability in sources, sinks, and atmospheric transport (see e.g., Monks et al. (2014) for a discussion on these processes). The diurnal profiles of $O_3$ are very different in different seasons (**Fig. 4 (b)**), with the largest day-night difference in summer and almost no difference in winter, owing to the increasing strength of photochemical production and the longer lifetime of $O_3$ respectively. The nighttime decrease in $O_3$ is greatest in summer and smaller during the other seasons, probably in large part due to deposition to the ocean surface.

### *2.3.2 Methane*

The long-term increase in the dry-air mixing ratio of $CH_4$ (Figure 5) is significant, 13 ppb/yr, and in line with observations made globally (e.g., Nisbet et al. 2019). The mean mixing ratio from the Atlantic wind sector (here roughly defined as wind direction between 210 and 260 degrees) is lower than the mean of all data, but shows the same trend. The overall mean $CH_4$ is about 0.02-0.03 ppm higher than the open-ocean mean.

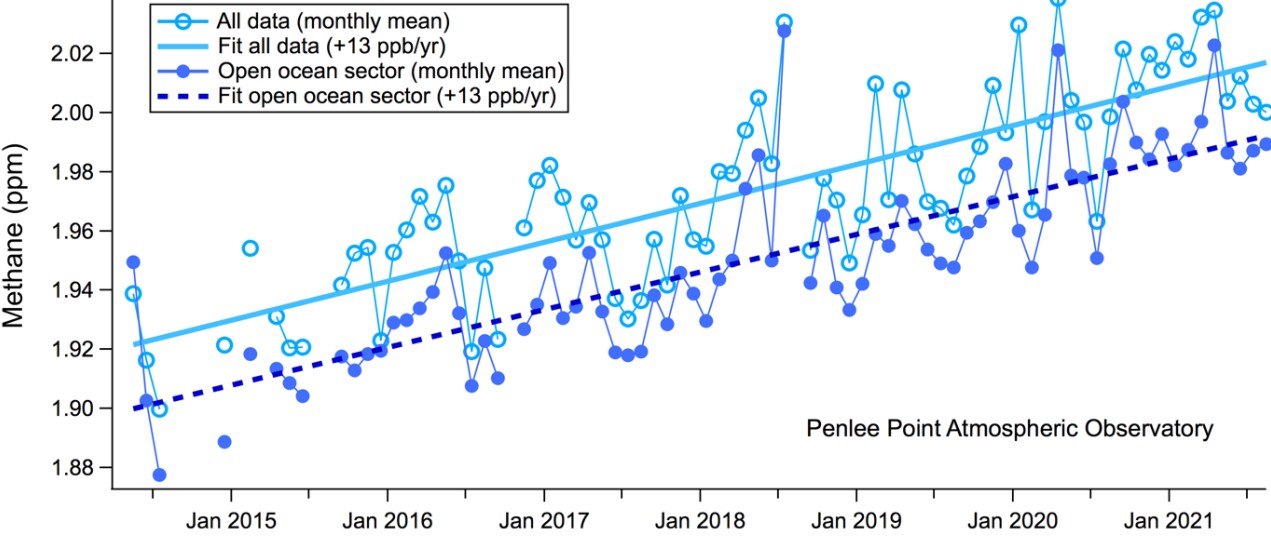

**Figure 5**: Long-term measurements of methane made at PPAO highlighting the strong increase in methane.

Methane shows a mean seasonal amplitude of 0.03 ppm (relative difference of 1.5%). The summer minimum is most likely due to increased sink of methane by the OH radical. These data suggest no significant deviation from the long-term trend over the last few years (2019-2022), when it has been postulated that the COVID lockdowns changed the atmospheric oxidising capacity and so the OH sink (e.g., Stevenson et al., 2022).

### *2.3.3 Aerosols from sunphotometers*

Long-term aerosol measurements (starting from 2001) have been made from the rooftop of PML (50.3661° N, 4.1482° W, about 10 km NNE of Penlee Point). The retrieved, cloud-filtered data are averaged to monthly intervals as shown in Figure 6. Overall there is no obvious long term trend in Aerosol Optical Depth (AOD) at this site, in contrast to many other locations in Western Europe that tend to show a gradual reduction. This may be because of the predominance of sea spray aerosols at this location (Yang et al. 2020).

The inferred size distributions are also shown (Fig. 6b). The volume distribution (dV/dlog(R)) is dominated by super-micron aerosols, while the number distribution (dN/dlog(R)) is dominated by sub-micron aerosols. There appears to be a gradual reduction in springtime aerosol maximum at around 100 nm radius from 2010 to 2021, which could be related to reduced terrestrial or ship anthropogenic emissions (e.g. due to air quality related regulations).

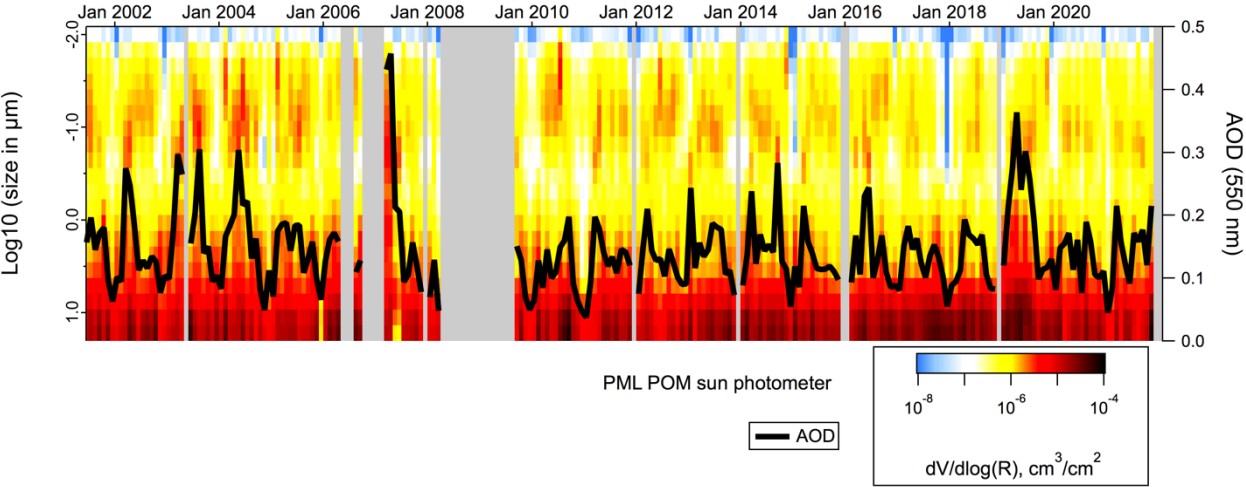

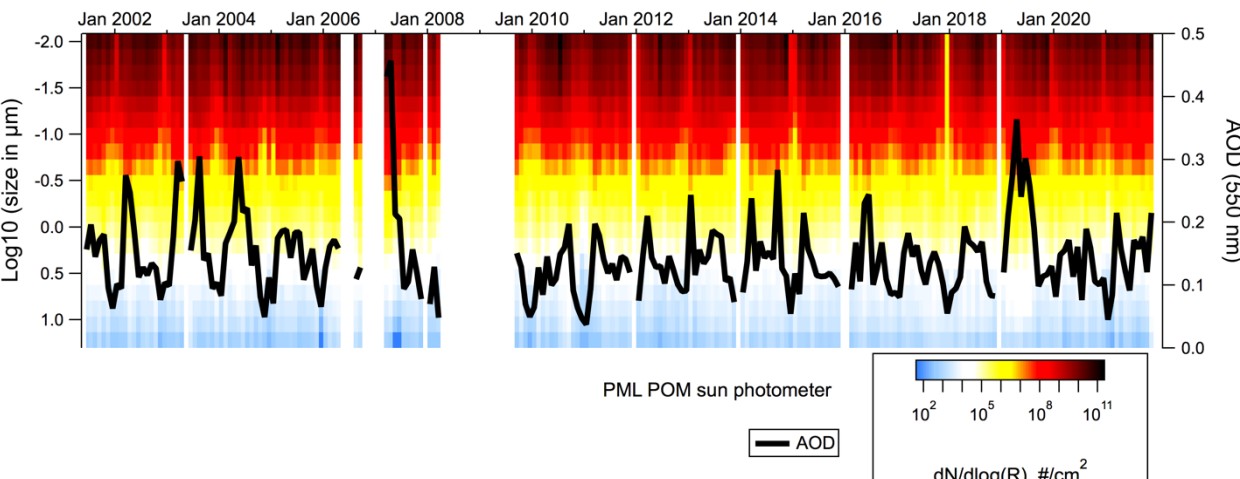

**Figure 6**. Long-term aerosol observations from the PML rooftop (monthly mean).

### 2.3.4 Data archive

Penlee Point Atmospheric Observatory data is archived at the CEDA: https://catalogue.ceda.ac.uk/uuid/8f1ff8ea77534e08b03983685990a9b0 (Plymouth Marine Laboratory and Yang (2024)). Data from the PML sun photometer can be found at https://dx.doi.org/10.5285/e74491c96ef24df29a9342a3d57b5939 (Smyth (2024)) The data format is ASCII, consisting of a header explaining the variables listed followed by the data in columnar format (one column per variable), with the data values in rows appearing in chronological order.



## 2.4 Atmospheric composition modelling with UKESM1

Model integrations were performed using a nudged (Telford et al., 2008) configuration of the UKESM1 Earth system model (Sellar et al., 2019) at Unified Model version 11.5. For nudged model integrations, the horizontal wind fields and potential temperature are relaxed to either the ERA-Interim (Dee et al., 2011) or ERA-5 (Hersbach et al., 2020) datasets using an e-folding relaxation timescale of 6 h. Sea-surface temperatures and sea-ice fields were prescribed from the Reynolds dataset (Reynolds et al., 2002). Atmospheric composition was simulated using the UKCA chemistry module, applying the stratosphere-troposphere chemical mechanism of Archibald et al. (2020) with the 2-moment prognostic aerosol scheme as described in Mulcahy et al. (2020). Simulations were performed from 1981 to 2014 using CMIP historical forcings (labelled as HIST) and continued until 2019 (ERA-Interim) or 2020 (ERA-5) using SSP3-7.0 forcings (labelled as SCEN) as per the AerChemMIP experiment definition (Collins et al., 2017) (see Table 6) for details.

In order to identify the impact of transport on modelled tropospheric ozone in the North Atlantic, the following diagnostic tracers were also defined:

- 4 different stratospheric ozone tracers (O3$_S$) were added. These are constrained in the stratosphere and evolve freely in the troposphere where they follow equivalent loss processes to the prognostic ozone field simulated by the model. The 4 O3$_S$ tracers are described below:
    1. Stratospheric concentrations are set to the prognostic ozone field above a model diagnosed tropopause defined by the 2PV+380K surface.
    2. Stratospheric concentrations are fixed at 1 ppmv above a model diagnosed tropopause defined by the 2PV+380K surface.
    3. Stratospheric concentrations are set to the prognostic ozone field above a model diagnosed tropopause defined by the WMO tropopause definition.
    4. Stratospheric concentrations are fixed at 1 ppmv above a model diagnosed tropopause defined by the WMO tropopause definition.

Tracer 1 and 3 are similar to the O3$_S$ tracers used in the CCMI experiments (Abalos et al., 2020) and represent tropospheric ozone originating from the stratosphere, while tracer 2 and 4 (also referred to as constant O3$_S$ tracers or O3$_{S-C}$) give a complementary measure of downward transport from the stratosphere that is not affected by stratospheric ozone geographical distribution or trends (Russo et al., 2023). An example of tracer 1 tropospheric column and its seasonal variation is given in Figure 7a.

- 30 regionally emitted tracers were included to diagnose long range transport into the North Atlantic region. These have either a lifetime of 5 or 30 days and emission regions are sketched in Figure 7b.



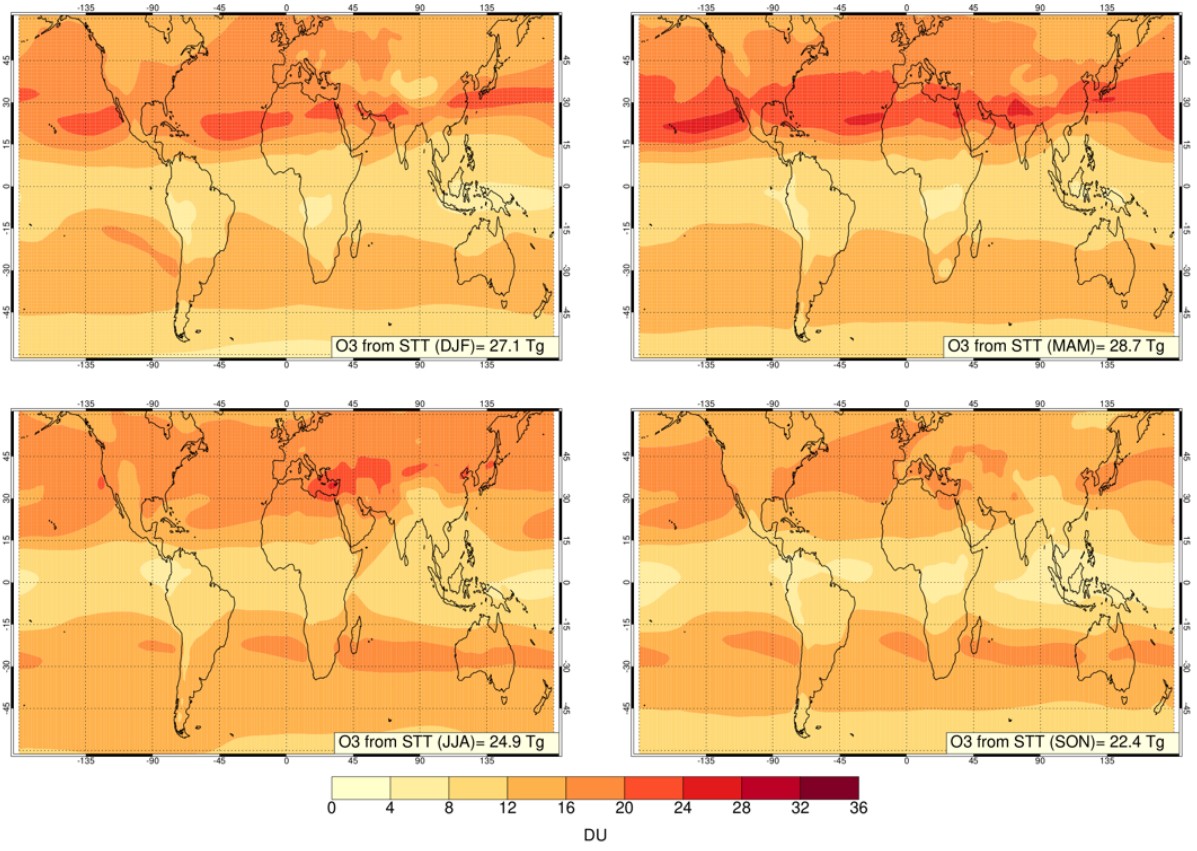

85

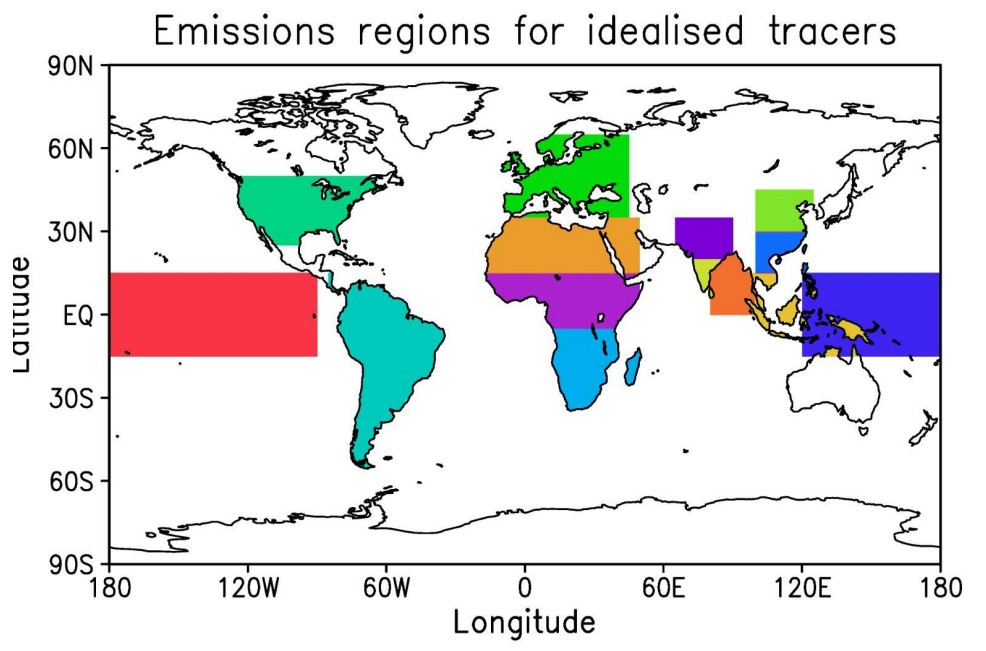

86





**Figure 7**. a) Integrated tropospheric column for the O3$_S$ tracer defined using prognostic ozone and the 2PV+380K tropopause, averaged over 2005-2017 using HIST1 and SCEN1 simulations (see Table 6 for details). b) Emission regions for the 5 day and 30 day regional tracers.

The simulations performed are listed in Table 6. The experimental design was focussed around providing simulations and output that could support observational campaigns and allowed for a detailed analysis of model transport and composition processes. As well as all the chemical and aerosol fields, fluxes through all chemical reactions and deposition processes were output as monthly means. Model restart files were also saved to allow for re-running short sections with an increased (and higher frequency) output request to compare against flight campaigns. Updates to the experiments were made throughout the project, incorporating bugfixes and model improvements.

**Table 6.** Description of the UKESM1 model simulations.

| Simulation | Nudging Dataset | Time Period | Notes | Rose suite ID |
|---|---|---|---|---|
| HIST1 | ERA-Interim | 1981-2014 | Settings as per UKESM1. | u-bv711 (01/1981-11/1991) and u-bw316 (12/1991-12/2014) |
| HIST2 | ERA-5 | 1982-2014 | Includes code-changes described in Ranjithkumar et al. (2021) | u-bw784 (01/1982-12-2014) |
| HIST3 | ERA-5 | 1982-2014 | Includes code-changes described in Ranjithkumar et al. (2021), technical improvements to the top-boundary condition of the tracers, updated photolysis rates, and the improved heterogeneous chemistry of Dennison et al. (2019) | u-bv828 (01/1982-05/2008) and u-bx320 (06/2008-12/2014) |
| SCEN1 | ERA-Interim | 2015-2019 | Continuation of HIST1 | u-by117 (SSP3-7.0) |
| SCEN2 | ERA-5 | 2015-2020 | Continuation of HIST2 | u-by803 (SSP3-7.0) |
| SCEN3 | ERA-5 | 2015-2020 | Continuation of HIST3 | u-by808 (SSP3-7.0) |

**2.4.1 Data archive**



892 Tb of UKESM1 model data were generated through the ACSIS project. A huge number of model diagnostics were output, including high time frequency fields (hourly) across the North Atlantic basin. These are listed here: https://www.ukca.ac.uk/wiki/index.php/ACSIS/u-bv711/STASH. Owing to the large nature of the model data set, selected core chemical species and tracers are available to download as monthly mean files from the CEDA dataset https://data.ceda.ac.uk/badc/acsis/UKESM1-hindcasts, Abraham (2024), these include ozone and ozone precursors (O$_3$, NO, NO$_2$, CO and methane) and the idealised tracers used to diagnose transport in the North Atlantic (four stratospheric tracers and thirty regionally emitted tracers). This data is available for all the model runs described in Table 6. The data is in Met Office PP format, which can be read using open access Python libraries held at https://ncas-cms.github.io/cf-python. If desired, users may also apply for a Met Office MASS (offline tape archive) account on the UK JASMIN data facility (https://jasmin.ac.uk) and search the Rose Suite IDs given in Table 6 for access to data from the specific experiments performed.

## 3 Ocean data sets

The North Atlantic Ocean is a major component of the overall North Atlantic Climate system and one of the key objectives of the ACSIS programme was to document the significant changes in ocean circulation and heat content which have taken place since the mid 20[th] century, to investigate the physical processes responsible and to identify their external drivers. Another objective was to understand how the ocean might change in the next several decades and to evaluate the potential impacts of these changes on human society and activities. In order to fulfil these objectives we compiled a substantial number of new data products and new model simulations.

The data products were compiled on the underlying principle of estimating components of the North Atlantic heat budget plus the sea surface temperature and sea surface height (dynamic and thermosteric) as these latter two are key to the wider impacts of the ocean on the atmosphere and on coastal sea level. Thus we bought together two basin scale observational estimates of the horizontal ocean volume and heat transports at 26°N described in previous publications (RAPID - https://rapid.ac.uk/rapidmoc/, McCarthy et al 2015; Moat et al., 2020) and at ~55°N (OSNAP - https://www.ukosnap.org/, Lozier et al., 2019), a new high spatial and temporal resolution Atlantic sea surface temperature dataset previously described by Williams and Berry (2020) and a new water mass preserving objectively interpolated ocean temperature and salinity dataset based on the international Argo float array described in Section 3.1 below(King, 2023) . On the modelling side, we undertook new cutting edge NEMO forced ocean model simulations with a variety of surface forcing datasets at resolutions of ¼° and 1/12°, described in Section 3.2.

Taken together, this new collection of model and observation based data has allowed us make significant advances in our understanding of North Atlantic variability including phenomena such as the impact of subpolar heat loss on the Atlantic Meridional Overturning Circulation (Megann et al., 2021a), the subpolar fingerprints of changes in the AMOC (Smeed et al.,



33 2018), , the origin of interannual changes in subpolar SST (Josey and Sinha 2022), the link between subpolar SST and European

34 winter weather (Grist et al., 2019) and summer heat waves (Mecking et al., 2019), the relationship between decadal variability

35 in surface and subsurface temperature (Moat et al., 2019) and the impact of subpolar freshwater input on the North Atlantic

36 atmosphere (Oltmans et al., 2020) to name just a few of the studies enacted under ACSIS.

**3.1 Ocean temperature and salinity, and upper ocean heat content**

As part of ACSIS the NOC has produced new ocean temperature and salinity datasets based on the Argo float array using

sophisticated optimal interpolation techniques which preserve ocean water masses. The dataset covers the period 2004-present

and extends to depths of up to 2000m. Two versions are available with spatial resolutions of 2 degrees and 1 degree

respectively. During ACSIS the main use of this dataset has been to calculate subtropical and subpolar heat content alongside

other available estimates in order to understand the interannual to decadal variability of the North Atlantic heat budget.

Here we illustrate the subpolar Ocean heat content (SOHC), which is an indicator of long-term changes in the heat supply to

the North Atlantic region (Figure 8). Changes in SOHC are thought to be important precursors of Atlantic Multidecadal

Variability (e.g. Sutton et al., 2018), and have been linked to changes in climate extremes, for example the number of Atlantic

hurricanes (Dunstone et al., 2011). The ACSIS SOHC time series are integrated from the region between 45°N to 67°N, and

80°W to 0E. The time series are calculated from gridded EN4.2.2 (Good et al., 2013) and Argo objectively mapped 1 x 1

degree temperature data sets (King, 2023). The SOHC calculated from the new dataset developed during ACSIS is shown in

red (based only on Argo measurements) while another calculation using the standard Met Office product EN4 (based on Argo,

hydrographic and remote sensing measurements) is shown in black. The two datasets agree well over the overlapping period

2004-present and the differences between the decadally filtered lines gives a useful indication of the uncertainty in the heat

content estimates due to the method of calculation.

**3.1.1 Data archive**

Objectively mapped temperature and salinity data and are available for download from BODC as self describing NetCDF

(http://doi.org/10.5065/D6H70CW6) files: https://doi.org/10.5285/fe8e524d-7f04-41f3-e053-6c86abc04d51 (King et al.,

2023) as are upper ocean heat content timeseries, also in NetCDF format :https://doi.org/10/g6wm, https://doi.org/10/g8g2

(Moat et al. (2021a-b)).

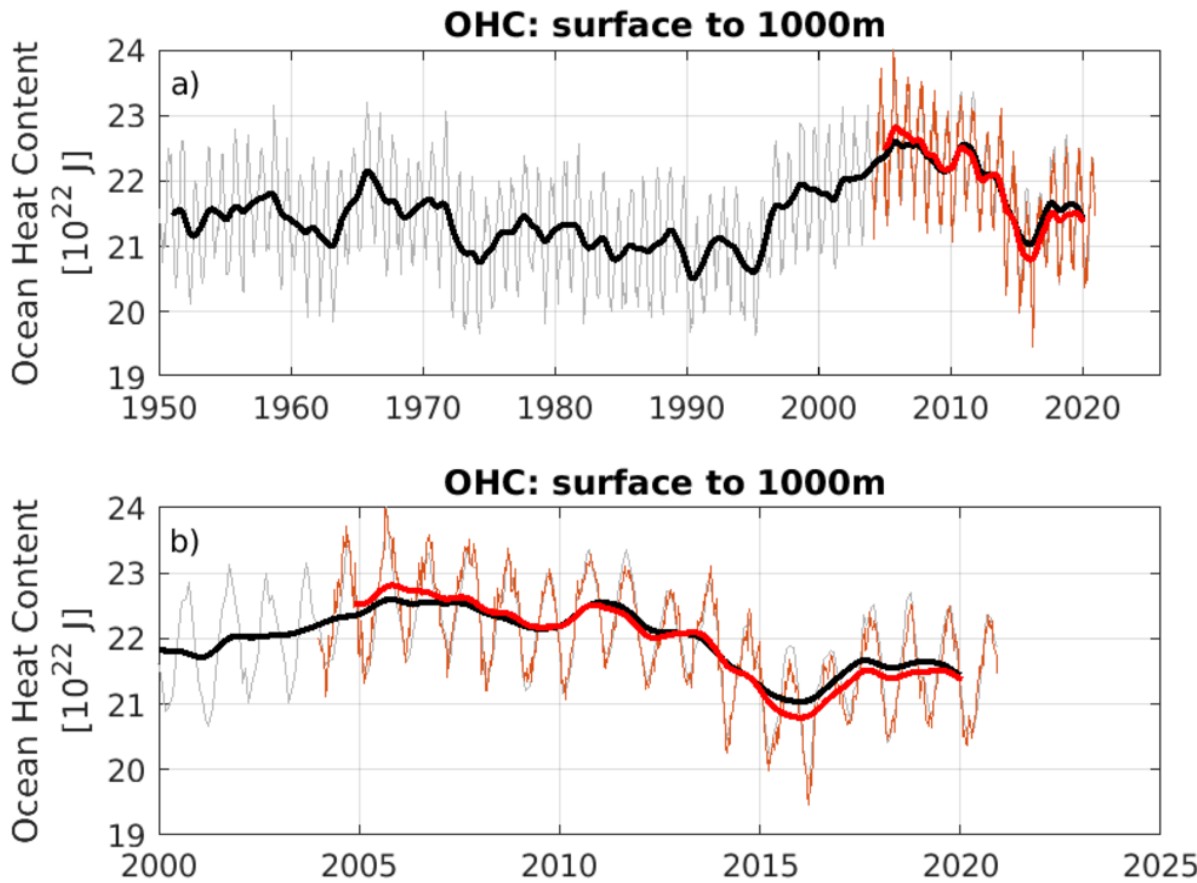

**Figure 8**. Subpolar ocean heat content index in units of $10^{22}$ J using EN4 (black) and ARGO OI (red)  a) 1950-2020 and b) during the Argo period 2004-2020). Thick lines have an annual low pass filter applied.

**3.2 Forced Ocean-ice simulations**

**3.2.1 ¼° ocean models forced with three different surface meteorological datasets.**

Three integrations of a global ocean and sea ice configuration, consisting of Global Ocean v6 (GO6, Storkey et al, 2018) and Global Sea Ice v8.1 (GSI8.1, Ridley et al, 2018) were carried out, as a deliverable for Work Package 2.3 of ACSIS, intended to provide a tool for scientific investigation of the mechanisms of variability of the AMOC and other modes of variability of the Atlantic Ocean. GO6 is based on NEMO v3.6 (Madec 2016), and GSI8.1 on CICE v5.2.1 (Hunke & Lipscomb, 2010; Ridley et al., 2018) The GO6 ocean configuration was chosen to be the same as that developed under the JMMP collaborative programme (http://www.jwcrp.org.uk/under/jmmp.asp) as the ocean component of the UK's submissions under CMIP6, namely GC3.1 (Williams et al.,2017) and UKESM1 (Sellar et al., 2019), and informed choices made in the UK OMIP (Ocean



Model Intercomparison Project – Griffies et al., 2016) integrations. Three forcing datasets were used to assess the sensitivity

of the models to the choice of forcing data. These were the CORE2 (Large and Yeager 2009), DFS5.2 (Brodeau et al 2010)

and JRA-55 (Tsujino et al., 2018) datasets, each supplying gridded surface met variables (air temperature, humidity, and

surface winds at subdaily intervals), surface radiative heat fluxes (downwelling shortwave and longwave at daily intervals)

and freshwater input (snow and precipitation at monthly intervals).

The simulations were run on a global domain on the eORCA025 1/4° grid, with 75 vertical levels. The integrations were run

from 1958 to 2007 (CORE2); from 1958 to 2015 (DFS5.2) and from 1958 to 2020 (JRA-55), and monthly means are archived.

Variables archived include full-depth potential temperature and salinity, horizontal and vertical velocity components, surface

fluxes of heat, freshwater and momentum; mixed-layer depth. Sea ice cover and thickness, but many other state and process

variables were also archived. Note that sea ice files from the JRA-forced run are only available for years 1990-2001 and 2002-

2020. These forced ocean-ice simulations use the same configuration as the ocean component of the coupled simulations

described in section 3.1.

A comparison of the model drifts in  globally averaged temperature and salinity is shown in figure 9. In particular there is a

large positive drift in upper ocean salinity in the DFS5.2 forced simulation and the relatively large freshening in the CORE2

simulations. Overall the JRA55 forced simulation shows moderate drift in both variables. Nonetheless simulated interannual

to multidecadal changes to Atlantic Ocean circulation are similar between the models (Fig 10). More details on the simulations

and the AMOC in the three simulations are given by Megann et al (2021a). We expect these simulations will be extremely

useful to investigate the role of surface forcing in generating model biases and in determining the mean ocean circulation and

its variability.

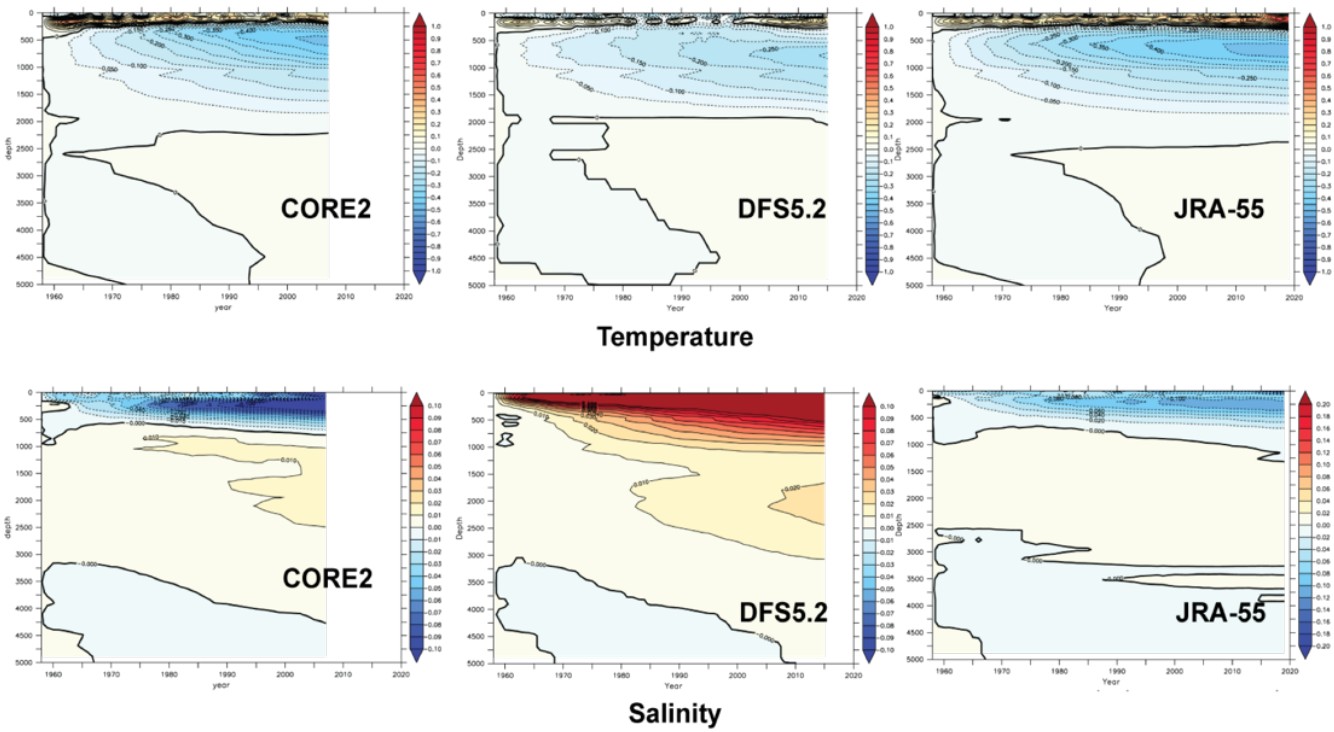

**Temperature**

**Salinity**

**Figure 9**. Annual drifts in global mean temperature (K) (top) and salinity (psu) (bottom) as a function of depth in the ACSIS ¼° forced ocean model simulations. Left panels are from the CORE2 forced simulation, centre panels are from the DFS5.2 forced simulation and right panels are from the JRA-55 forced simulation.

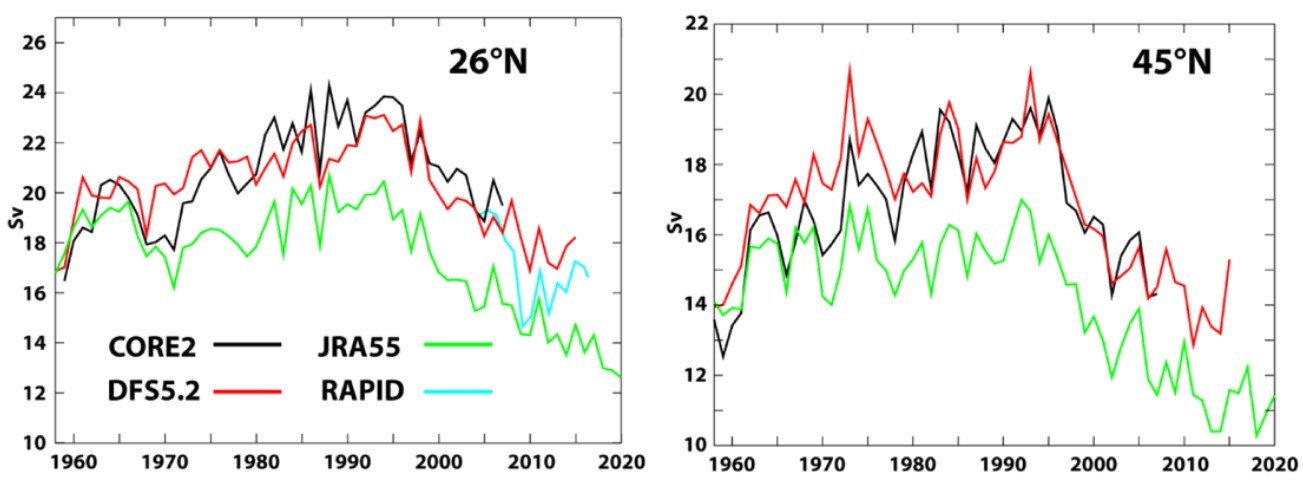

AMOC time series in ACSIS GO6 integrations at (left) 26°N and (right) 45°N.



**Figure 10**. AMOC timeseries (Sv), 1960-2020 from the ACSIS ¼° forced ocean model simulations at 26°N (left) and 45°N (right). Timeseries from all three integrations are shown on each panel: CORE2 forced simulation (black); DFS5.2 forced simulation (red) and JRA-55 forced simulation (green). The AMOC derived from observations at 26°N (the RAPID-MOCHA array), available from 2004 onwards, are plotted on the left panel (cyan).

### 3.2.2 ¼° and 1/12° "twin" simulations

Two integrations of the Global Ocean v8p7 (GO8p7) ocean and sea ice configuration simulation were run under the ACSIS programme. This is based on NEMO v4.0.4 (Madec et al., 2019), including the SI3 sea ice model, and has been developed under the Joint Marine Modelling Programme (JMMP see http://www.jwcrp.org.uk/under/jmmp.asp). The simulations are identical apart from the ocean horizontal resolution: one on a ¼° grid, and the other a 1/12° grid. They are forced with the JRA-55 surface forcing dataset (Tsujino et al, 2018) from 1958 to 2021. The integrations are intended to provide a tool for scientific investigation of the mechanisms of variability of the AMOC and ocean heat content of the Atlantic Ocean at an eddy-rich resolution. The GO8p7 configuration is close to that expected to be incorporated in the GC5.1 coupled climate model and the UKESM2 earth system model, both aimed at CMIP7. The configuration was implemented at the two resolutions, with the parameter and physics setting as close as possible (there are some necessary changes to lateral friction which are required for numerical stability at higher resolution), to investigate the sensitivity of the circulation, numerical mixing and other metrics to the resolution.

As for section 4.4.1 The integrations were carried out on a global domain on eORCA025 1/4° and eORCA12 1/12° grids, with 75 vertical levels. The integrations were run from 1958 to 2020 and monthly and annual means of the 3-D and 2-D model fields were saved (including full-depth potential temperature and salinity, horizontal and vertical velocity components, surface fluxes of heat, freshwater and momentum; mixed-layer depth, and sea ice cover and thickness). 5-day means of a selection of surface fields (including SST, mixed layer depth and sea-surface height) are also archived.

To illustrate the simulations we show timeseries of some key globally integrated variables from the twin simulations and also, for context, from the three ¼° simulations already described in section 4.4.1 (Fig 11). Global mean temperature drifts are of order 0.05K over the ~50 year integrations or 0.001K yr$^{-1}$. The 1/12° simulation has a smaller drift than its twin ¼ degree resolution. The twin simulations show positive temperature drift while the other simulations how a negative drift. We expect to see an SST warming trend under the influence of anthropogenic warming superimposed on interannual and decadal variability. All the simulations show strong interannual variability with about the same amplitude and timing, forced by interannual changes in wind stress and buoyancy forcing, and not influenced by global temperature and salinity drifts. On decadal and longer timescales the difference between variability, secular trends and model drifts can be blurred. The models all show a small reduction in global mean SST from initialisation to the late 1970s. The DFS5.2 forced simulation then continues to reduce its SST until the mid 1980s after which the SST remains more or less stable until about 2010, however all the other simulations increase their SST at a fairly steady rate throughout the 1980s, 90s and 2000s. from about 2010 onwards



all the simulations experience strong surface warming. Globally integrated downward surface heat flux is consistent with the

global mean surface temperature evolution with a negative net surface flux in the early decades for the three simulations with

different surface flux forcing and a positive net flux for the twin simulations. The net heat flux for the twin simulations is

generally positive whereas for the other simulations it only becomes positive around the year 2000 and this is when the global

mean temperature in those simulations starts to rise. The downward heat flux clearly shows the signals of large volcano

eruptions (Agung, 1964, el Chichon 1982 and Pinatubo 1991) as well as the 1997 El Nino event (see Balmaseda et al 2013).

The sharp downward dip in 2009 is interesting and possibly linked to the sudden AMOC reduction at that time, but further

research is required to investigate this. With the exception of the DFS5.2 forced simulations, global mean salinity and global

mean surface salinity show quite small trends consistent with a reasonably balanced surface freshwater flux.  The DFS5.2

forced simulation shows strong salinification consistent with a net loss of freshwater through the surface. The twin runs show

best conservation of freshwater. Finally, the net heating/cooling and freshening/salinification of the simulations is reflected in

the global mean sea surface height which is most stable in the twin simulations.

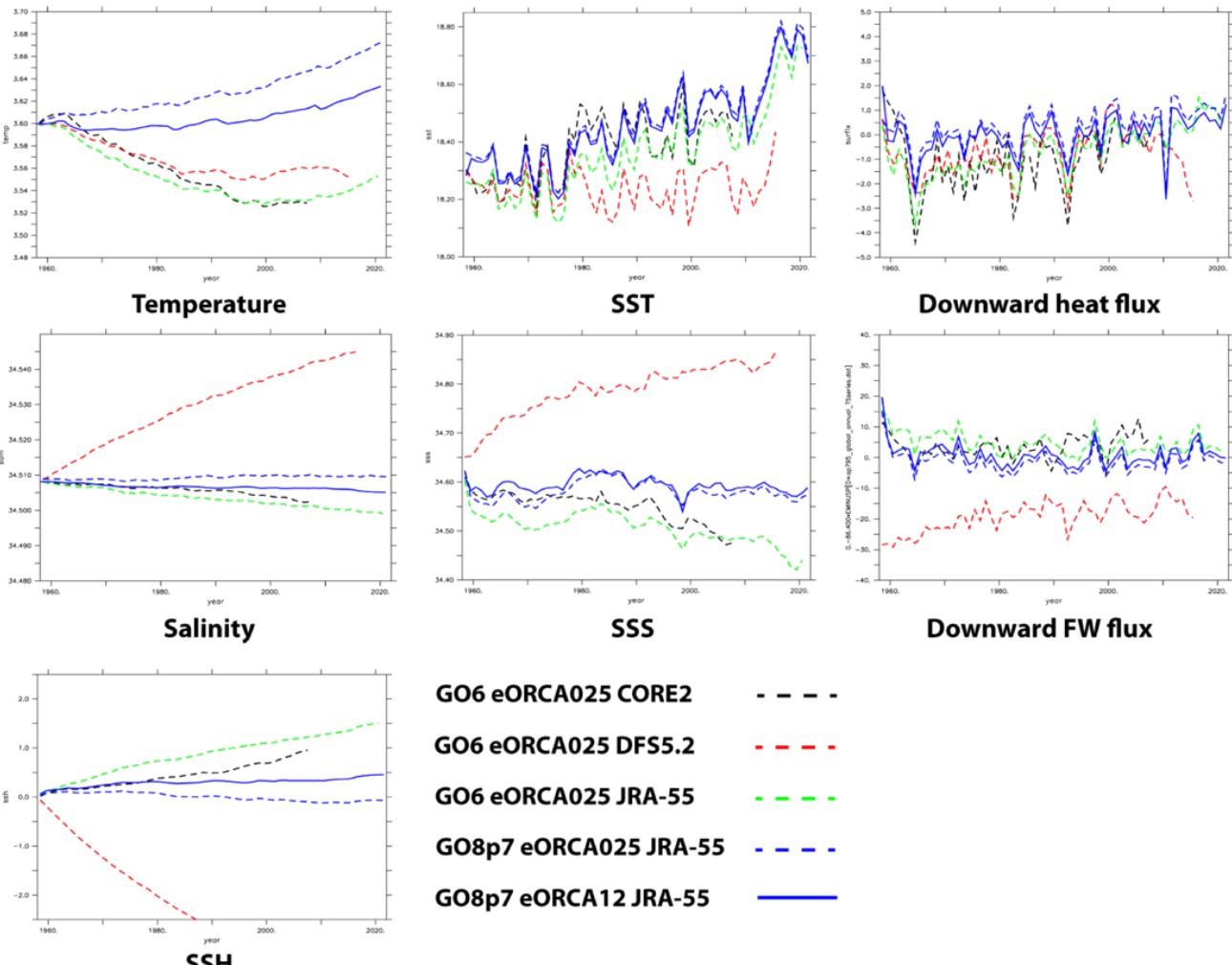

**Figure 11**. Trends in key variables in the ACSIS ¼° and ½° forced ocean simulations. The variables plotted are global mean temperature (top left), global mean sea-surface temperature (top centre), global mean net downward air-sea heat flux (top right), global mean salinity (second row left), global mean sea-surface salinity (second row centre), downward freshwater flux (second row right) and global mean sea-surface height (bottom left). Dashed lines are from the ¼ degree model (CORE2 forced – black, DFS5.2 forced – red, JRA-55 forced, ¼° twin simulation – blue) whilst the solid blue line is from the 1/12° twin simulation. Note that the green and blue lines are both from JRA-55 forced model simulations but with different model code versions and configurations (see text).

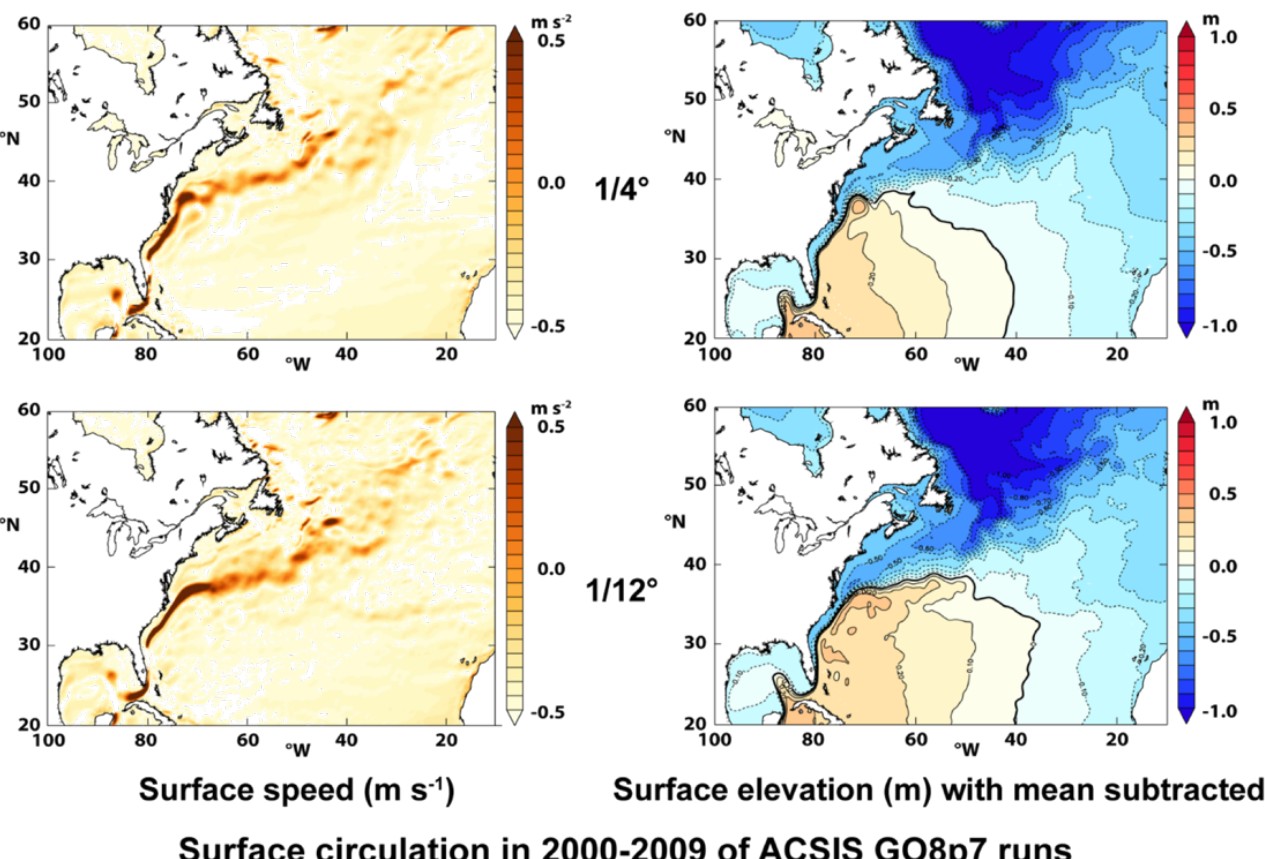

**Figure 12**. Surface North Atlantic circulation from the ACSIS twin simulations averaged over years 2000-2009. Anomalous surface current magnitude (m s$^{-2}$) for the ¼° simulation (top left) and for the 1/12° simulation (bottom left), Anomalous mean sea surface height (m) for the ¼° simulation (top right) and the 1/12° simulation (bottom right).

A final illustration shows the mean surface circulation in the North Atlantic from the twin simulations (Fig 12). The most obvious difference in the surface velocity (left hand panels) is that the Gulf Stream separation is more realistic in the 1/12° simulation where the current moves northeastwards off Cape Hatteras (~38°N). This contrasts with the ¼° simulation where the current shifts direction anticlockwise to remain quite close to the coast. The kink in the Gulf Stream Extension at the Northwest corner (~50°W, 40°N) is also more realistic in the 1/12° simulation and there is also a discernible signature of the Azores current (zonal feature around 34°N) which is missing in the ¼° simulation. Similar features can be seen in the mean sea surface height from the two simulations (right panels). One interesting difference is in the penetration of the Labrador Current much further south in the 1/12° simulation – where the low sea surface heights characteristic of the subpolar gyre penetrate south west along the North American shelf/slope region north of the Gulf stream extension (between 80°W and 50°W and 35°N to 45°N). Decadal variability in the position of the Gulf Stream has been shown to be linked to salinity anomalies



that are advected southwards by the Labrador Current (New et al., 2022) so these differences between the simulations are

likely to impact on their simulation of AMOC variability.

*3.2.3 Data archive*

Data from all the ocean simulations are archived in NetCDF format, with separate files for variables defined on the T, U, V

and W grids (one for each month of simulation) as is standard for NEMO. Each variable has a long name which gives a detailed

description of the variable (see Madec, 2016, 2019 for an explanation of the data output format). Separate monthly NetCDF

files contain sea ice variables and lagrangian iceberg properties trajectories on the CICE grid. The data are archived at CEDA

(Megann et al., 2021b, c, d):

CORE2-forced run: https://dx.doi.org/10.5285/119a5d4795c94d2e94f610647640edc0 (Megann et al., 2021b,

DFS5.2-forced run: https://dx.doi.org/10.5285/a0708d25b4fc44c5ab1b06e12fef2f2e,(Megann et al., 2021c)

JRA55-forced run: https://dx.doi.org/10.5285/4c545155dfd145a1b02a5d0e577ae37d (Megann et al., 2021d)

¼° "twin" simulation: https://dx.doi.org/10.5285/e02c8424657846468c1ff3a5acd0b1ab (Megann et al., 2022a)

1/12° "twin" simulation: https://dx.doi.org/10.5285/399b0f762a004657a411a9ea7203493a (Megann et al., 2022b).

**4 Ice data sets.**

**4.1 Advanced Sea Ice model simulations**

Results from 6 forced ocean-ice simulations and 2 stand-alone ice simulations are included to document the impact of sea ice

physics and atmospheric forcing data on the Arctic sea ice evolution. All of them use the same sea ice model CICE

configuration GSI8.1 (Ridley et al., 2018) and the ocean-ice simulations use the same ocean model NEMO GO6.0 (Storkey et

al., 2018) as the forced ocean ice simulations of section 4.4 and the  HadGEM3 climate model of sections 3.1. Three different

atmospheric forcing data set are applied: NCEP Reanalysis-2 (NCEP2) data (Kanamitsu et al., 2002, updated 2020), CORE2

surface data (Large & Yeager, 2009) and the atmospheric forcing data set DFS5.2 (Dussin et al., 2016). Regarding the sea ice

component, we use the default CICE setup as in HadGEM3 (CICE-default) and an advanced setup (CICE-best) in which a

new process is added (snow loss due to drifting snow) and some adjustments have been made to model physics and parameters.

See Schroeder et al. (2019) and Table 7 for details.

**Table 7**. Overview of model simulations with default and improved sea ice processes.

| Simulation | Atmospheric forcing | Ocean model | CICE setup | Time period |
|---|---|---|---|---|





| CICE-default | NCEP2 | Mixed-layer | CICEv5.1.2 with prognostic melt pond model and EAP rheology | 1980-2020 |
|---|---|---|---|---|
| CICE-best | NCEP2 | Mixed-layer | As CICE-default, but with several modifications including snow drift scheme, bubbly conductivity scheme, increased sea ice emissivity and reduced melt pond max fraction parameter (see Schroeder et al., 2019) | 1980-2020 |
| NEMO-CICE-1deg-default-CORE | CORE II | NEMOv3.6 | CICEv5.1.2 with prognostic melt pond model | 1960-2009 |
| NEMO-CICE-1deg-best-CORE | CORE II | NEMOv3.6 | As CICE-best | 1960-2009 |
| NEMO-CICE-1deg-best-DFS | DFS5.2 | NEMOv3.6 | As CICE-best | 1960-2015 |
| NEMO-CICE-1deg-best-NCEP | NCEP2 | NEMOv3.6 | As CICE-best | 2000-2020 |
| NEMO-CICE-1/4deg-default-DFS | DFS5.2 | NEMOv3.6 | CICEv5.1.2 with prognostic melt pond model | 1979-2015 |
| NEMO-CICE-1/4deg-best-DFS | DFS5.2 | NEMOv3.6 | As CICE-best, but with increased ice and snow conductivity instead of snow drift scheme | 1979-2015 |

The impact of our changes to the sea ice model on the fidelity of the model sea ice simulation is shown in Figure 13. All simulations with the default CICE setup (thin lines) underestimate the mean Arctic sea ice thickness during winter. Figure 13 shows that the mean Arctic Cyrosat-2 sea ice thickness is more than 50cm thicker in April than in those simulations. By applying the advanced CICE setup, all simulations (stand-alone, NEMO-CICE 1° and NEMO-CICE 1/4°, thick lines) show realistic mean April sea ice thickness. The advanced setup leads to improvements in simulating summer sea ice extent, too (not shown) and highlights the importance of sea ice physics for accurate model simulations for the Arctic.

### 4.1.1 Data archive

Data from the global ocean simulations with advanced sea ice are archived in NetCDF format as described in section 3.2.3 above. Standalone sea ice simulations are similar but output consist of a single NetCDF file containing sea ice variables on the CICE grid for each month of simulation. The data is accessible via CEDA: http://catalogue.ceda.ac.uk/uuid/770a885a8bc34d51ad71e87ef346d6a8 (see Megann et al., 2021e).

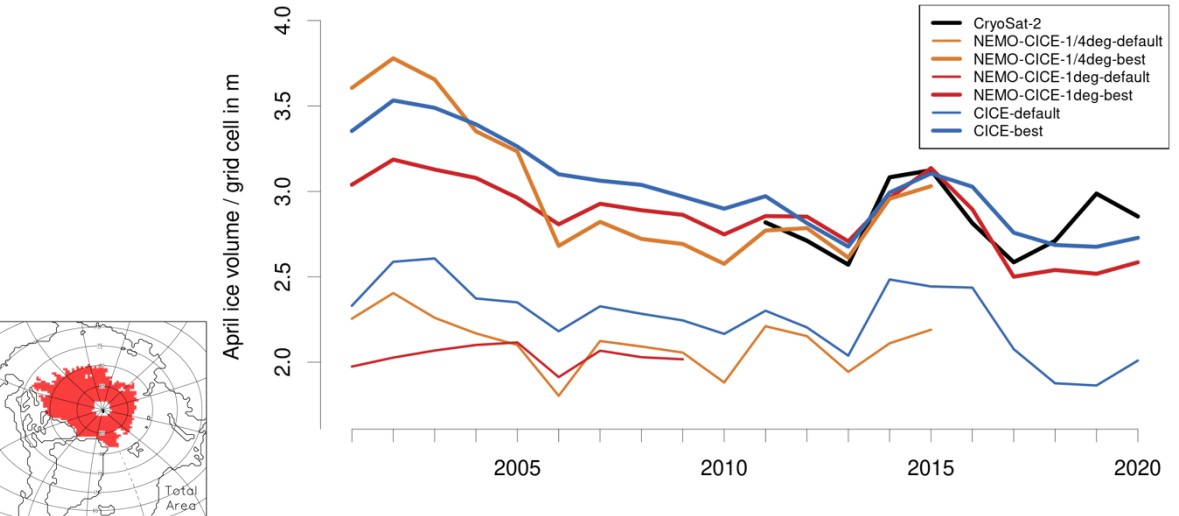

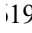

**Figure 13.** Mean April Arctic Sea ice volume over red region for several model simulations in comparison to Cyrosat-2 estimates. The selected region represents the area over which Cryosat-2 data are available for the whole period from 2010 to 2020 (October to April). Table 7 provides more information about the setup of the model simulations.

## 5. Previously published ACSIS datasets

The new datasets described in the previous sections should be viewed in the context of (and potentially used in conjunction with) several other datasets generated in whole or in part by the ACSIS programme and already published and described in the scientific literature. Here we provide a brief overview of these other datasets and include links to where they can be accessed.

### 5.1 Stratospheric Aerosol Surface Area Density from Explosive Volcanic Eruptions

The "MajorVolc" UM-UKCA volcanic aerosol datasets are model simulations of the monthly progression of the volcanic aerosol clouds from the 3 largest volcanic eruptions of the 20th century – 1963 Agung, 1982 El Chichon and 1991 Pinatubo. The interactive stratospheric aerosol model experiments break new ground, being an advance on the 2D global aerosol model simulations that generated the CMIP6 volcanic aerosol dataset (Arfeuille et al., 2013; Luo, 2016), the stratospheric circulation and dynamics progressing in 3D, within the high-top N96L85 GA4 UM-UKCA composition-climate model (Walters et al., 2014). The simulations apply the v8.2 of the GLOMAP-mode aerosol microphysics module (Mann et al., 2010; Dhomse et al., 2014; Mann et al., 2015) including recent adaptations for the stratosphere (Brooke et al., 2017; Dhomse et al., 2020). This upgraded capability predicts the volcanic forcings with very high fidelity, each of the steps in the formation, growth and sedimentation of the sulphuric acid aerosol particles, the UM-UKCA model also calculating the oxidation rate of the volcanic $SO_2$, consistently with the depletion and replenishment of oxidants that occurs for the amount of $SO_2$ emitted.



Within ACSIS, the volcanic forcings were published as monthly-varying global 2D zonal-mean datasets for 4 key aerosol properties (stratospheric AOD, surface area density, particle effective radius, aerosol extinction at 550nm & 1020nm) on an open-access data archive (Dhomse, 2020). The variables align directly with Figures in the peer-reviewed journal article (Dhomse et al., 2020), and retain the 2 key parent resolutions of the model (1.25˚ latitude vs 1 km altitude to 40km).

Following the protocols for the co-ordinated Historical Eruption $SO_2$ Emission Assessment (HErSEA) experiment within the international ISA-MIP activity for interactive stratospheric aerosol models (Timmreck et al., 2018), three different datasets were produced for each eruption, at upper-limit, lower-limit and mid-range emission of $SO_2$ (e.g. 10, 14, 20 Tg of $SO_2$ for Pinatubo). For each $SO_2$ emission amount, 3 ensemble members were run, each initialised from a timeslice control run whose stratosphere was spun-up for the decade's GHG & ODS loadings, re-start points chosen to enact the QBO phase transition for that eruption (see Dhomse et al., 2020).

Monthly mean volcanic forcing data is stored in netCDF format. Files are archived for each variable, all 3 ensemble members for each "eruption realisation" (SO2 emission amount) within 1 file (e.g. "Pin10Tg_saod_2Ms_mon.nc"). The dataset identifier is https://doi.org/10.17632/n3g2htz9hk.1 (Dhomse (2020)).

Whereas the MajorVolc volcanic aerosol datasets provide globally gridded aerosol properties such as Surface Area Density and aerosol extinction, in order to enact volcanic forcing in a climate model requires one to specify the aerosol optical properties across the solar and terrestrial spectral ranges: mapping extinction, absorption and asymmetry parameters ($Q_{ext}$, $Q_{abs}$, $g$) to the wavebands of the radiative transfer module within the climate model. This was done for the Major Volc datasets generated as part of ACSIS, described above, and these data are available as described in Feng et al. (2021) and Dhomse et al. (2021a, b):

Pinatubo (https://doi.org/10.5281/zenodo.4739170 (Feng et al., 2021)); El Chichon (https://doi.org/10.5281/zenodo.4744633 (Dhomse et al., 2021a)); Agung (https://doi.org/10.5281/zenodo.4744686 (Dhomse et al., 2021b)).

**5.2 CMIP6 HighResMIP global climate model simulations**

The 6th Coupled Model Intercomparison Project (CMIP6) HighResMIP (https://www.highresmip.org/) sub project aimed to increase the atmosphere and ocean resolution of global climate models to at least 50 km in the atmosphere and 0.25° in the ocean, and to assess the effect of these increases in resolution on process representation and model fidelity (Haarsma et al. 2016, Roberts et al. 2018). The UK contribution to CMIP6 HighResMIP, based on the HadGEM3 climate model (Hewitt et al 2011), was delivered as part of the EU Horizon 2020 PRIMAVERA project (https://www.primavera-h2020.eu/). Some of the HadGEM3 PRIMAVERA simulations were co-funded by ACSIS and are referenced here (Table 8). These consisted of



atmosphere only simulations with horizontal resolutions of N256 (~50km) and N512 (~25km) (Table 8, rows 1-12), and analogous fully coupled simulations with an ocean resolution of 0.25 degrees (Table 8, rows 13-24). The simulations were conducted in pairs consisting of a historical simulation from 1950-2014 and a future simulation from 2015-2050. The terminology is detailed in Haarsma et al. (2016) and Roberts et al. (2019). All these simulations are available on the Earth System Grid Federation (ESGF, https://esgf.llnl.gov/). Output is in CF-compliant NetCDF for all simulations. The NEMO ocean component in these simulations is the same configuration as the forced ocean model simulations described in section 3.2.

Two further cutting edge simulations were performed at even higher resolution in both ocean and atmosphere, 1/12°, and 50km (N512) respectively (Table 8, rows 25-26). The first was a control 1950s climate running from 1950-2014 and the second was a future simulation (SSP5-8.5) from 2015-2050. See Roberts et al., (2020) for an assessment of the simulated Atlantic Meridional Overturning Circulation in this and other HigResMIP simulations.

**Table 8**. Summary of HighResMIP global climate model simulations. The first 12 rows refer to SST-forced atmosphere only simulations, the remaining rows refer to coupled ocean-atmosphere simulations.

| Model | Experiment | Resolution (Atm./Ocn.) | Period | Ensemble member | DOI |
|---|---|---|---|---|---|
| HadGEM3-GC31-MM | highresSST-present | N216 | 1950-2014 | r1i1p1f1 | http://doi.org/10.22033/ESGF/CMIP6.6029 Roberds (2017a) |
| HadGEM3-GC31-MM | highresSST-future | N216 | 2015-2050 | r1i1p1f1 | http://doi.org/10.22033/ESGF/CMIP6.6013 Roberts (2019a) |
| HadGEM3-GC31-MM | highresSST-present | N216 | 1950-2014 | r1i2p1f1 | http://doi.org/10.22033/ESGF/CMIP6.6029 Roberts (2017a) |
| HadGEM3-GC31-MM | highresSST-future | N216 | 2015-2050 | r1i2p1f1 | http://doi.org/10.22033/ESGF/CMIP6.6013 Roberts (2019a) |
| HadGEM3-GC31-MM | highresSST-present | N216 | 1950-2014 | r1i3p1f1 | http://doi.org/10.22033/ESGF/CMIP6.6029 Roberts (2017a) |
| HadGEM3-GC31-MM | highresSST-future | N216 | 2015-2050 | r1i3p1f1 | http://doi.org/10.22033/ESGF/CMIP6.6013 Roberts (2019a) |
| HadGEM3-GC31-HM | highresSST-present | N512 | 1950-2014 | r1i1p1f1 | http://doi.org/10.22033/ESGF/CMIP6.6024 Roberts (2017b) |
| HadGEM3-GC31-HM | highresSST-future | N512 | 2015-2050 | r1i1p1f1 | http://doi.org/10.22033/ESGF/CMIP6.6008 Roberts (2019b) |





| | | | | | |
|---|---|---|---|---|---|
| HadGEM3-GC31-HM | highresSST-present | N512 | 1950-2014 | r1i2p1f1 | http://doi.org/10.22033/ESGF/CMIP6.6024 Roberts (2017b) |
| HadGEM3-GC31-HM | highresSST-future | N512 | 2015-2050 | r1i2p1f1 | http://doi.org/10.22033/ESGF/CMIP6.6008 Roberts (2019b) |
| HadGEM3-GC31-HM | highresSST-present | N512 | 1950-2014 | r1i3p1f1 | http://doi.org/10.22033/ESGF/CMIP6.6024 Roberts (2017b) |
| HadGEM3-GC31-HM | highresSST-future | N512 | 2015-2050 | r1i3p1f1 | http://doi.org/10.22033/ESGF/CMIP6.6008 Roberts (2019b) |
| HadGEM3-GC31-HM | hist-1950 | N512, 0.25° | 1950-2014 | r1i1p1f1 | http://doi.org/10.22033/ESGF/CMIP6.6040 Roberts (2018a) |
| HadGEM3-GC31-HM | highres-future | N512, 0.25° | 2015-2050 | r1i1p1f1 | http://doi.org/10.22033/ESGF/CMIP6.5984 Roberts (2019c) |
| HadGEM3-GC31-HM | hist-1950 | N512, 0.25° | 1950-2014 | r1i2p1f1 | http://doi.org/10.22033/ESGF/CMIP6.6041 Schiemann et al. (2019a) |
| HadGEM3-GC31-HM | highres-future | N512, 0.25° | 2015-2050 | r1i2p1f1 | http://doi.org/10.22033/ESGF/CMIP6.5985 Schiemann et al. (2019b) |
| HadGEM3-GC31-HM | hist-1950 | N512, 0.25° | 1950-2014 | r1i3p1f1 | http://doi.org/10.22033/ESGF/CMIP6.6040 Roberts et al. (2018a) |
| HadGEM3-GC31-HM | highres-future | N512, 0.25° | 2015-2050 | r1i3p1f1 | http://doi.org/10.22033/ESGF/CMIP6.5984 Roberts et al 2019c |
| HadGEM3-GC31-HH | hist-1950 | N512, 1/12° | 1950-2014 | r1i1p1f1 | https://doi.org/10.22033/ESGF/CMIP6.5881 Roberts (2018b) |
| HadGEM3-GC31-HH | highres-future | N512, 1/12° | 2015-2050 | r1i1p1f1 | https://doi.org/10.22033/ESGF/CMIP6.1822 Coward and Roberts (2018) |



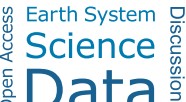

'89

**5.3 Sea surface temperature**

'90

'91  The ACSIS Atlantic Ocean medium resolution SST dataset is a 5-day field of Sea Surface Temperature (SST) on a ½° by ½°

'92  grid from 1950 to 2014 and covers the Atlantic Ocean (http://dx.doi.org/10.5285/83b0cd7e7cc6495a90b4cb967ead3577,

'93  Williams and Berry (2020)).

'94  The dataset is based on *in situ* ship and buoy SST observations from the International Comprehensive Ocean-Atmosphere Data

'95  Set (ICOADS) Revision 3. Measurements which fail initial quality control checks were rejected and for each grid box where

'96  there is data a trimmed mean and sample standard deviation was calculated to produce super-observations. These were then

'97  expressed as anomalies from the 1981-2014 Climatology (mean, annual, semi-annual and tri-annual) from the European Space

'98  Agency (ESA) Climate Change Initiative (CCI) SST dataset (version 2.0) derived from satellite observations. The

'99  measurements were then interpolated using Kriging to infill gaps and estimate uncertainties. The spatial covariance used in

'00  the Kriging was derived from the CCI SST analysis residuals (CCI SST analysis minus the CCI SST climatology). After

'01  interpolation, bias corrections derived from the HadSST.4.0.0.0 dataset are applied.

'02  The dataset is available as annual CF complaint NetCDF files, with a total of 65 annual files available. Each file contains: the

'03  5 day mean sea surface temperature; the corresponding climatological value, the sea surface temperature anomaly and the

'04  uncertainty in the sea surface temperature.

'05  The new dataset has been developed as part of the ACSIS for use in validation and comparison with regional climate models.

'06  Other potential uses include boundary forcing for regional reanalyses, monitoring and assessment of regional climate change

'07  and other studies requiring SST at a resolution higher than typical for *in situ* products (i.e. < 1 month, < 1°) and spanning the

'08  satellite and pre-satellite era. In Figure 14a we compare the new SST dataset with two other leading SST datasets over the

'09  whole Atlantic domain and find good agreement of the basin averaged variability and trends. Fig 14b shows the data in a

'10  barcode type plot emphasising Atlantic climate warming. Additionally Figure 14b shows a first principal component time

'11  series from an EOF analysis of the three timeseries.

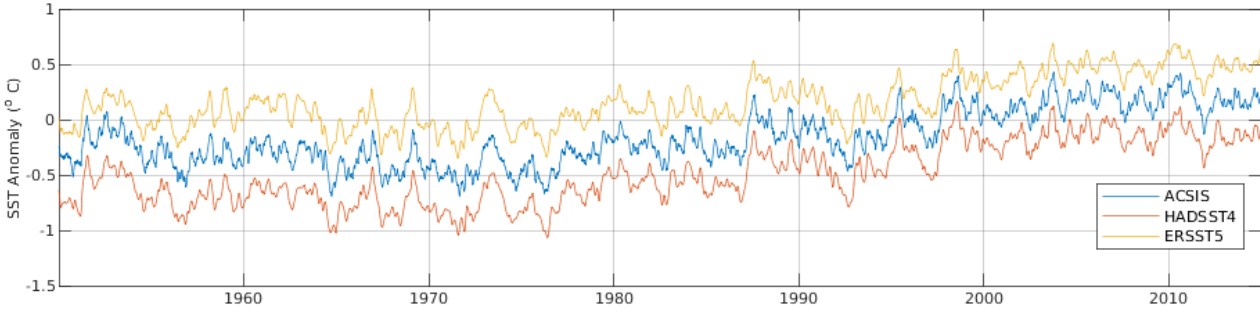

'12

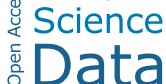

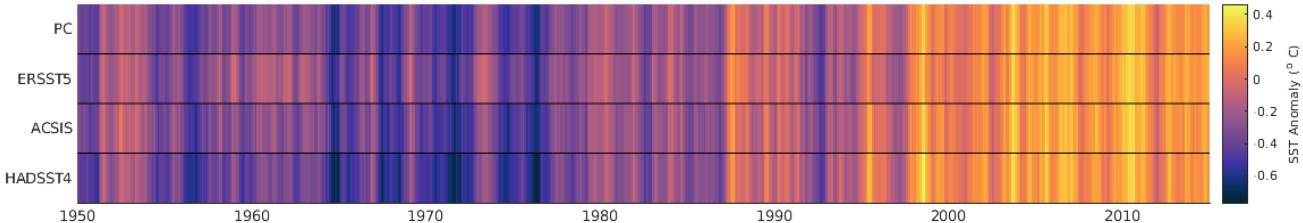

'13

'14    **Figure 14**. (a) SST anomalies time series averaged over the whole Atlantic (60°S to 68°N and 98°W to 20°E) of the new

'15    ACSIS dataset (blue) compared two leading global SST datasets, HadSST4 (Kennedy et al., 2019) and ERSST5 (Huang et al.,

'16    2017). (b) The three datasets as a barcode plot. A timeseries of the principal component of the EOF of the three SST timeseries

'17    is plotted at the top of the barcode. SST anomalies are relative to a 1981-2014 climatology.

'18

'19    **5.4 Atlantic Meridional Overturning Circulation (AMOC) observed at 26.5°N**

'20

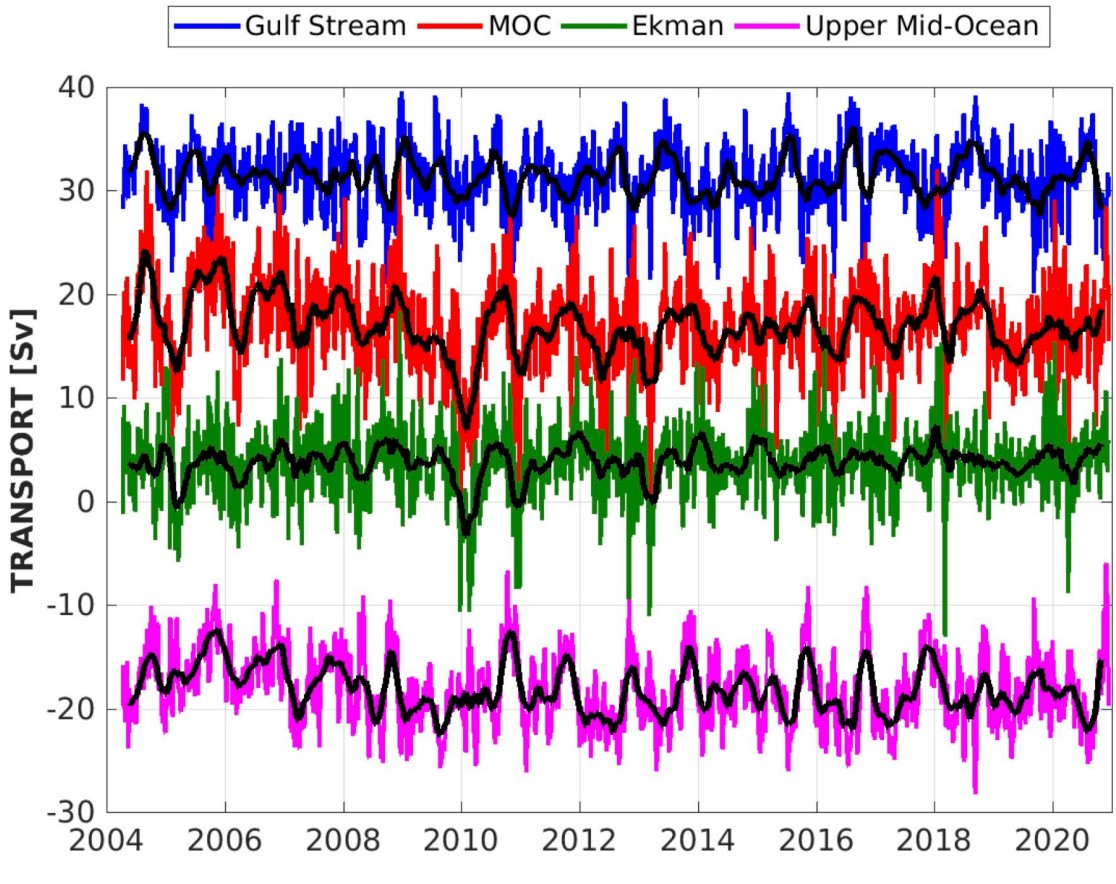

'21



'22

'23 **Figure 15**. The Atlantic Meridional Overturning Circulation (AMOC) index (red) and its components the Gulf Stream/Florida

'24 Strait transport (blue), the Ekman transport (green) and the upper mid-ocean transport (magenta) at 26.5°N derived from the

'25 RAPID array observations. Units are in Sverdrups (Sv) where $1Sv = 10^6 \ m^3 s^{-1}$. Coloured lines are 10-day values while black

'26 lines have had a three month filter applied. The AMOC index is the maximum of the AMOC streamfunction which is defined

'27 as the zonal integral of the area weighted velocity, summed downwards with depth from the surface. The depth of the maximum

'28 is variable but generally lies in the depth range 700-1000m (McCarthy et al., 2015).

'29

'30 The AMOC is an iconic index of North Atlantic ocean circulation and numerous observational and modelling studies have

'31 established the importance of its associated northwards heat and salt transport to the mean climate and interannual-decadal

'32 variability of the subpolar North Atlantic region (Robson et al., 2018, Moat et al., 2019), to regional climate elsewhere on the

'33 globe (Monerie et al., 2018), and to global climate via its connection to the global thermohaline circulation. Whilst the RAPID

'34 observations have been supported by separate funding, ACSIS provided essential support for the processing and analysis of

'35 RAPID observations. ACSIS scientists have analysed many important phenomena uncovered by the RAPID array, including

'36 the origin, impact and predictability of the extreme AMOC reduction in 2010 (Germe et al., 2022), the transition to a weaker

'37 AMOC after 2008 (Smeed et al., 2018), possible AMOC recovery from 2009 (Moat et al., 2020), and its subsequent impact

'38 on ocean conditions in the eastern subpolar gyre (Bryden et al., 2020). Figure 15 shows the entire timeseries from 2004 to the

'39 present day including the separately measured components due to the near surface Ekman transport (green), the transport

'40 through the Florida Straits (blue) and the geostrophic transport east of the Bahamas (magenta). The net transport (red) clearly

'41 shows the reduction in magnitude since 2008 and the major downturn in 2010. Note that at the time of writing the AMOC has

'42 yet to recover to its pre 2008 value. AMOC time series data can be downloaded directly from the RAPID project website

'43 (https://rapid.ac.uk/rapidmoc, Moat et al., 2022)

'44

'45 **5.5 Ice observations**

'46 Pan-Arctic sea ice thickness is estimated using satellite data from ESA's CryoSat-2 (CS2) mission. Launched in 2010, CryoSat-

'47 2's main payload is a Ku-band radar altimeter (SIRAL), which measures the elevation of Earth's surface. Sea ice freeboard

'48 (the portion of an ice floe above the waterline) is measured by differencing the elevation of the sea ice floe and that of the

'49 surrounding ocean. Sea ice freeboard is then converted to thickness by assuming that sea ice floats in hydrostatic equilibrium

'50 in the ocean, and assuming values for snow depth, and snow, ice and ocean density. CryoSat-2's orbit repeats every ~30 days,

'51 providing Arctic-wide sea ice thickness estimates every month from October-April. The method and dataset are detailed in

'52 full in Tilling et al., (2018), and monthly sea ice thickness, gridded at 5km, are available from the CPOM data portal

'53 http://www.cpom.ucl.ac.uk/csopr/seaice.php.

'54



'55    For the purposes of the ACSIS project, we binned individual CryoSat-2 sea ice thickness estimates provided by CPOM into

'56    the five default ice thickness categories of the sea ice model CICE on a rectangular 50 km grid: (1) ice thickness $h<0.6$ m, (2)

'57    0.6 m $<h<1.4$ m, (3) 1.4 m $<h<2.4$ m, (4) 2.4 m $<h<3.6$ m, and (5) $h>3.6$ m (Schroeder et al, 2019). The mean area fraction and

'58    mean thickness are then derived for each thickness category. One of the key motivations of binning the CS2 along-track data

'59    into sub-grid ice thickness classes is to assess the role of the ice thickness distribution (ITD) in model initialisation and to

'60    quantify the realism of the CS2 ITD against independent estimates from airborne data. In addition to the bespoke data described

'61    above, monthly (October-April, 2010-2021) 5km-gridded sea ice thickness estimates are available (in ASCII and NetCDF

'62    formats) on the CPOM data portal: http://www.cpom.ucl.ac.uk/csopr/seaice.php.

'63

'64    **6 Summary**

'65    We have described the multidisciplinary model and observational datasets that were produced by the UK ACSIS programme

'66    and how and where the data can be accessed. The scope of ACSIS was very broad, covering atmospheric composition,

'67    atmospheric circulation, ocean circulation, ice sheets, sea ice, and their interactions, and this breadth is reflected in the rich

'68    variety of datasets generated. We note that whilst the focus of the ACSIS programme was the North Atlantic, most of the

'69    model products covered the global domain, and many of the observational products have both global and regional significance.

'70    Despite its great size and scope, the ACSIS programme had finite resources and so was not able to fully exploit the data it

'71    generated. The landmark ACSIS papers cited here can be seen as starting points for further research. Therefore we believe

'72    there is a major opportunity to repurpose our data for new research studies to build on the substantial financial and intellectual

'73    investment that ACSIS represents and we express the hope that the ACSIS datasets provide a lasting legacy to the international

'74    environmental science community.

'75

'76    **Appendix A: Overview of select aircraft composition instruments**

'77    ***UoM Time of Flight Chemical Ionisation Mass Spectrometer***

'78    The University of Manchester High Resolution-Time of Flight-Chemical Ionisation Mass Spectrometer (ToF-CIMS),been

'79    described in detail by Matthews et al., (2013) for aircraft deployment. Briefly, iodide ions cluster with sample gases in the ion-

'80    molecule reaction region (IMR) region creating a stable adduct. The flow is then sampled through a critical orifice into the

'81    first of the four differentially pumped chambers in the TOF-CIMS, the short segmented quadrupole (SSQ).  Quadrupole ion

'82    guides transmit the ions through these stages. The ions are then subsequently pulsed into the drift region of the ToF-CIMS

'83    where the arrival time is detected with a pair of microchannel plate detectors with an average mass resolution of 4000 (m/Δm).

'84    The inlet design is an atmospheric pressure, rearward facing, short residence time inlet, consisting of 3/8" diameter

'85    polytetrafluoroethylene (PTFE) tubing with a total length to the instrument of 48 cm. A constant flow of 12 SLM is mass flow

'86    controlled to the ion-molecule reaction region (IMR) using a rotary vane pump (Picolino VTE-3). 1 SLM is then subsampled

'87    into the IMR for measurement.

'88



An Iris system as described by Lee et al. (2018) was employed to pressurise and mass flow control the sample flow into the instrument, avoiding sensitivity changes that would be associated with variations in pressure inflight that is not controlled sufficiently by the constant flow inlet. This works upon the principle of the manipulation of the size of the critical orifice in response to changes in the IMR pressure. As with the Lee et al., (2018) design, this works by having a stainless steel plate with a critical orifice and a movable PTFE plate on top of this, also with a critical orifice. These orifices either align fully and allow maximum flow into the instrument or misalign to reduce flow. This movement is controlled by the 24VDC output of the IMR Pirani pressure gauge in relation to the set point and was designed collaboratively with Aerodyne Research Inc. The IMR set point was 72±3 mbar for the aircraft campaigns which is set through a combination of pumping capacity on the region (Agilent IDP3), mass flow controlled reagent ion flow and sample flow. The reagent ion flow is 1 SLM of ultra-high purity (UHP) nitrogen mixed with 2 SCCM of a pressurised known concentration gas mix of CH3I in nitrogen, passed through the radioactive source, 210Po. The total flow through the IMR is measured (MKS MFM) at the exhaust of the Agilent IDP3 pump so that not only is the IMR pressure monitored but also the sample flow. All mass flow controllers and mass flow meters are measured and controlled using the standard Aerodyne Inc EyeOn control unit and software.

A pressure controller is also employed on the short segmented quadrupole (SSQ) region to make subtle adjustments in this region independently of any small IMR changes that may occur inflight. This works upon the principle controlling an electrically actuated solenoid valve in a feedback loop with the SSQ pressure gauge to actively control a leak of air into the SSQ pumping line. The SSQ is pumped using an Ebara PDV 250 pump and held at 1.8±0.01 mbar.

Instrument backgrounds are programmatically run for 6 seconds every minute for the entire flight, by overflowing the inlet with ultra high purity (UHP) nitrogen at the point of entry into the IMR. Here a 1/16th inch PTFE line enters through the movable PTFE top plate, ensuring that the flow exceeds that of the sample flow. Inlet backgrounds are also run multiple times during campaigns manually by overflowing as close to the end of the inlet as possible with UHP nitrogen. Data is taken at 4Hz during a flight, which is routinely averaged to 1 Hz for analysis. Of the 6 points in each background, the first 2 and last point are unused and the mean of the background is calculated using custom python scripting. Backgrounds are humidity corrected and using linear interpolation, a time series of the instrument background is determined and then subtracted to give the final time series (Matthews, 2023).

### UoM Aerosol Mass Spectrometer

The chemical composition of non-refractory submicron aerosols (organic (OA), sulphate, nitrate, ammonium and non-sea-salt chloride) can be measured by a compact time-of-flight Aerosol Mass Spectrometer (C-ToF-AMS, Aerodyne Research Inc, Billerica, MA, USA) (Drewnick et al., 2005), which provides chemical characterization across a range of ion mass-to-charge (m/z) ratios from 10 to 500. The detailed operation of the AMS, including calibration and correction factors, during aircraft deployment has been described previously (Morgan et al., 2009). In brief, aerosols enter the instrument via an aerodynamic



lens inlet, focusing the incoming particles into a narrow beam. The aerodynamic lens system of the AMS in this study is tailored to sample submicron aerosols. Particles exit the aerodynamic lens into the particle-sizing chamber, which is evacuated to progressively lower pressures as the particle beam passes through and removes the majority of the gaseous material. Non-refractory components of the particles are then flash vaporised on a resistively heated porous tungsten surface. The resultant gaseous molecules are ionised by a 70-eV electron beam released from a tungsten filament. These fragment ions are analysed by a Time-of-Flight mass spectrometer (ToF-MS). The AMS mass spectra were recorded every 8 or 15 s during the ACSIS campaign (ACSIS-1 and 3-6). The AMS data was processed using the standard SQUIRREL (SeQUential Igor data RetRiEvaL, v.1.65C) ToF-AMS software package. The AMS data was also calibrated using monodisperse ammonium nitrate and ammonium sulfate particles. A time- and composition-dependent collection efficiency (CE) was applied to the data based on the algorithm by Middlebrook et al. (2012).

### *UoY LIF-SO2*

The University of York LIF-$SO_2$ instrument is a custom-built system for the highly sensitive detection of $SO_2$ via laser-induced fluorescence, and is based on the system originally demonstrated by Rollins et al. 2016. The basic operating principle is the excitation of $SO_2$ at 216.9 nm, generated from the fifth harmonic of a custom-built tuneable fibre-amplified semiconductor diode laser system at 1084.5 nm, and the subsequent detection of the resultant fluorescence photons. The laser wavelength is rapidly (~10 Hz) tuned on and off a strong $SO_2$ transition, with the difference between these signals being directly proportional to the $SO_2$ concentration within the sample cell. The laser wavelength is tracked using a reference cell containing a known $SO_2$ concentration.

The ACSIS-7 experiment was part of the first field deployment for the York LIF-$SO_2$, and was thus in part a learning experience on the operation of the instrument aboard an aircraft. The sample flow rate was maintained at 2 slpm and the use of a ram inlet allowed both the sample and reference cells to be operated at 400 mbar for the full altitude range of the campaign to maximise instrument sensitivity. Multi-point calibrations were carried out across the expected concentration range approximately every half an hour to ensure the instrument sensitivity was well characterised. To assess the possible quenching effect of excited $SO_2$ by water vapour, or increased wall losses when sampling humid air, calibrations in both stable ambient air and dry zero air were carried out, for which this effect proved negligible. The uncertainty in the LIF-$SO_2$ measurements was calculated predominantly from the uncertainty in the instrument sensitivity (typically 6 %). However, due to inconsistencies in the laser power and laser linewidth, the sensitivity was seen to vary during the course of each flight. Therefore, a mean sensitivity has been applied and this variation has been conservatively added to the sensitivity uncertainty on a flight-by-flight basis to give an overall uncertainty of ~ 15 % (using the mean of this variation). The 3 σ precision of 225 ppt has also been determined conservatively from stable ambient measurements due to issues with completely overflowing the instrument inlet with zero air in flight.



**Code/Data availability**

Code availability is not applicable for this article. All data is deposited in reliable data repositories and access is detailed in Table 1 of this article.

**Author contributions**

ATA and BS prepared the original draft with input from TJB, LJC, EM, KR, MRR, FAS, KR, LT, LW, HW, MY

BS, EM and MRR edited the original draft, all authors reviewed the manuscript.

SJJB, TJB, EM, CR. FAS, LT, NT, LW, HW acquired data.

ATA, LJC, HC, PE, JL, BS, MY, acquired funding

**Competing interests**

There are no competing interests.

**Acknowledgements**

We gratefully acknowledge the financial support provided by the UK Natural Environment Research Council for the extensive data provided by the ACSIS project. Airborne data were obtained using the BAe-146 Atmospheric Research Aircraft flown by Airtask Ltd and managed by FAAM Airborne Laboratory, jointly operated by UK Research and Innovation and the University of Leeds. We would like to give special thanks to the Airtask pilots and engineers and all staff at FAAM Airborne Laboratory for their hard work in helping plan and execute successful flight campaigns during ACSIS. PE and LT were supported by NERC awards NE/T008555/1 and NE/S007458/1 for the development and operation of the LIF-SO2. MY, TB, and the Penlee Point Atmospheric Observatory measurements were supported by the NERC projects ACSIS (NE/N018044/1) and MOYA (NE/N015932/1). TS and the Plymouth sunphotometer measurements were supported by the NERC project ACRUISE (NE/S005390/1) and by the Western Channel Observatory, which is funded by NERC through its National Capability Long-term Single Centre Science Programme, Climate Linked Atlantic Sector Science (NE/R015953/1). We further thank Frances Hopkins, Jani Pewter, Daniel Phillips, and Simone Louw for instrument maintenance at Penlee Point Atmospheric Observatory. We thank Luis Neves, Instituto Nacional de Meteorologia e Geofísica, São Vicente (INMG), Mindelo, Cabo Verde and, Shalini Punjabi, WACL, for technical assistance in the CVAO measurements. Model simulations were performed at NCAS, NOC and CPOM under ACSIS grants NE/N018001/1 and NE/N018044/1.



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
