# Peer review of "Data supporting the North Atlantic Climate System: Integrated Studies (ACSIS) programme,"

_Earth System Science Data, 2023_

## Author Comment (AC1)

Many thanks to the Editor and the Reviewers for their efforts with our manuscript. Here are our responses (in highlighted italics)

Reviewer 1

Data supporting the North Atlantic Climate System: Integrated Studies (ACSIS) programme, including atmospheric composition, oceanographic and sea ice observations (2016-2022) and output from ocean, atmosphere, land and sea-ice models (1950-2050)

by A. T. Archibald et al.

Summary: This is a very heterogeneous data set publications wherein the authors try to assemble a suite of physical and chemical observations that in one way or the other have to do with the ACSIS programme. The wealth of the data sets mentioned and described is immense. Overall, I find that the authors solved the challenge to group the different data sets into thematic containers quite well. Naturally, the diversity of the data sets and sources means that the manuscript contains quite a number of tables but this is ok given the heterogeneity of the data sets mentioned. Also, the paper provides a convincing list of links to respective respositories where the data, which are partly a subset of larger collectons, can be accessed. Overall, I find this manuscript useful. I have a number of concerns, though, that I would ask the authors to think about and iterate the manuscript accordingly - as specified in my general comments and also in my specific comments.

General Comments:

GC1: This data set publication contains - to my taste - a too large amount of results from pre-liminary data set analysis that is better to be put into other publications. There are quite some paragraphs in this manuscript as written, which read like an advertizement of the many things that have already been done with the data and/or that could potentially be done with data. I do not find this appropriate for a data set publication of this kind. You find more about this in my specific comments.

*We have toned down much of the preliminary analysis (especially the trend analyses) and the citations of work done in ACSIS as highlighted in the specific comments by both reviewers. Instead, we have tried to confine ourselves to comments which will highlight any limitations of the data or which will aid in their interpretation.*

GC2: At the same time, the manuscript - as written - is overly light when it comes to detail data set quality, reliability and limitations of use. While I understand that the majority of the data sets mentioned here stem from in situ observations, more emphasis on uncertainty sources and potentially limited reliability of the data obtained would immensely assist users in doing a good job when utilizing your data in their research.

*Thank you for the comment, we have done our best to include as much additional information as possible (e.g. Table 2 includes information on precision and uncertainty, we go to some effort to show the origin of the air sampled by the Penlee Point observatory, and model simulations are all ensembles, sampling uncertainty in surface fluxes, model resolution, ice physics etc). We are however ready to respond to any specific lack of clarity or uncertainty that the Reviewer is able to point to.*

GC3: Section 5 contains a - to me somewhat unmotivated - addition of other data sets that may or may not be already published in the context of ACSIS. This I find sub-optimal and I suggest to only keep those data set descriptions of section 5 that immediately have to do with the suite of data that is published with this data set publication AND put the respective information either into a separate section after the introduction or to split section 5 and include the respective information into the specific subsections to which these auxiliary data directly contribute. See my specific comment in this regard

*While we understand the Reviewer's reasoning, we would prefer to keep this section. In the pre review stage the material now in section 5 was scattered throughout the other sections, but the editor advised that already published material should be gathered together into a separate section. In terms of justifying the inclusion of the material, we note that each of the subsections relates to one of the earlier sections. Thus, the stratospheric aerosol subsection (5.1) relates to the atmospheric composition (Section 2), the HighResMIP simulations subsection (5.2) relate to the ocean model simulations (Section 3), the AMOC and SST subsections (5.3 and 5.4) relate to the ocean observations (Section 3) and ice observations subsection relates to the sea ice modeling (Section 4). The Reviewer has already noted the relevance of the ice subsection. We similarly think that the subpolar heat content timeseries included in Section 3 should be used in conjunction with or in the context of concurrent SST and AMOC observations as it is generally accepted that these three diagnostics are a tightly coupled triad. Again, we think it would be unhelpful not to mention the existence of numerous coupled counterpart simulations using essentially the same model configuration (domain and parameter settings) as the forced ocean-ice simulations presented in Section 3). So we argue that these three subsections will be very useful for potential users of the new datasets. Finally, the volcanic stratospheric aerosol datasets presented in subsection 5.1 are perhaps slightly less relevant but do provide some context for the nudged historical simulations described in Section 2.4 which extend from the 1980s to the 2010s and so include the El Chichon and Pinatubo eruptions.*

GC4: Quite a number of the figures and illustrations need further work to make the content readable; a few errors need to be corrected. See my specific and editoral comments in this regard. *OK*

Specific Comments:

L115: Maybe consider to have such a footprint map in this data science paper as well?

*A new schematic map has been added (Figure 1 of the new manuscript)*

L161 / Table 2: There are two Ozone measurement instruments that seem to have been used in succession. How was the inter-instrument calibration carried out to ensure that the ozone measurements of the two instruments are comparable?

*The Thermo Fisher Scientific Model 49i and 2B Technologies Model 205 dual beam photometers operated onboard the FAAM Airborne Laboratory are routinely calibrated against an ozone primary standard (Thermo Fisher Scientific model 49i-PS) maintained by the University of York's COZI laboratory, part of the Wolfson Atmospheric Chemistry Laboratories. This primary standard maintains the traceability of ozone photometers operated by ground-based and airborne facilities of the National Centre for Atmospheric Science to NIST, using the UK National Physical Laboratory's Standard Reference Photometer No 20.*

*The two photometers operated by FAAM have been test-flown alongside each other, sharing a common inlet, to ensure measurement comparability.*

*The figure below illustrates such a test flight demonstrating excellent agreement for the full altitudinal range of the FAAM research aircraft (35000 ft).*

[Figure]

*The following figure illustrates the linear fit of ozone mixing ratios recorded by the two photometers, averaged to 4 seconds (the Model 49i measurement rate) and weighted with uncertainties (3% or 3 ppb), for 28 missions flown by FAAM between 02/07/2019 and 14/02/2020.*

[Figure]

*In some occasions, the ozone comparison between both measurements did not fall within combined uncertainties as shown in the following figure.*

[Figure]

*We have investigated this bias in our laboratories in 2020 and found it is related to the behaviour of the Dewlines/Nafion™ water exchange membrane technology employed in the 2B photometer with changing humidity. This issue has also been identified by Eric Hintsa at the NOAA Global Monitoring Laboratory (https://doi.org/10.5194/amt-14-6795-2021), who fly the UCATS payload also utilising 2B Technologies photometers. Our experience and further discussions with 2B Technologies concluded that the performance water exchange membrane degrade over time under certain conditions.*

*The potential bias is accounted for in our increased ozone measurement uncertainty for the 2B Technologies model 205, namely ±5 ppb or (3%+2 ppb) whichever is greater.*

*A full description of the 2B Technologies Model 205 ozone photometer can be found in this Zenodo reference https://doi.org/10.5281/zenodo.7503437 published 04/01/2023.*

L190: What is the scientific rationale to perform this grouping? Isn't the density of measurements changing a lot with altitude so that a linear grouping is perhaps not optimal?

*Figure 2 has now been updated so that the heights of the boxes represent the number of observations within each altitude bin (taller box corresponds to more observations). ACSIS flights were planned so that measurements were taken as evenly as possible through the atmosphere (within the range of the aircraft). The 1000 m altitude grouping was chosen as a compromise between showing structural detail in the atmosphere and readability of the plot.*

Figure 2:

- I understand that all six panels show a mixing ratio. However, it would increase the readability of the panels if you'd remove the titles and instead include the respective information into the x-axis caption, i.e. "NO mixing ratio [ppt]" or "CO mixing ratio [ppb]".

*Changed and new figure added to MS.*

- Another area of improvement would be to use the same y-axis scaling. Currently these differ between the top four panels and the bottom two panels.

Changed.

- Also, I count nine bins in the bottom two panels but ten bins in the top four panels. None of these fit to the noted 1000 m bin chosen.

*Measurements of O3, CO, CO2, CH4, were made up to ~9300 m (text has been updated to give correct altitude range now). The NOx instrument cannot give reliable data above 27000 ft/ ~8200 m and so there is no data presented for the top bin range (9000 m – 10000 m).*

- While the bottom two panels seem to fit to the maximum measurement altitude of 7600 m mentioned in the text, the top four panels do not. Please correct the panels and/or correct also the text since it does not fit to these panels.

*This was an error in the text and line 167 been changed to reflect the true maximum altitude observations were made at.*

- Does the vertical line shown in the boxes denote the mean or the median value?

*Median value. Figure caption has been updated to reflect this.*

- Does the horizontal extent of the boxes denote the interquartile range, i.e. from 0.25 to 0.75?

*Yes, and the figure caption has been updated to reflect this.*

- Do the bars extending left and right from the box denote 1, 2, or 3 standard deviations?

*The bars extend to the smallest and largest values no more than 1.5 times the interquartile range. The caption has been updated to reflect this.*

L193-195: I am not sure I understand correctly. Has the filtering that is mentioned here been done only for the sake of improving the readability of the panels shown? Or were data excluded from the data set? If the latter, what is the scientific rationale to exclude those high mixing ratios?

*The filtering has been done to improve the readability of the panels and the figure caption text has been updated to explain this.*

L198-204:

- I am not convinced that the provided ascii files (one kind with comma separation, one kind

with empty space separation) are an overly user-friendly access point to the wealth of data that is going to be published here. Making this data available in netCDF file format would be substantially more useful. The authors might want to motivate why they decided to not make an effort to provide the data in both, ascii text and netCDF file format.

*We have provided the data in both ascii and netcdf formats as described in section 2.1.4: "The merged files are open access and designed to be a tool for an initial exploration of the data and to highlight the breadth of the atmospheric composition data collected during the ACSIS programme. However, for further analysis the original frequency data should be used and details of where these files can be found is included in the header information of the merged files"*

- Please check this paragraph for "merge file" vs. "merged file" [correct], and also check for punctuations; one is missing in L203 while one is too much in L204.

*We found a few inconsistencies and have now used "merged" throughout the manuscript.*

L228+++ (all subsections 2.2.1-2.2.4)

- I am not sure I understand why this paper, dedicated to be a data set publication, comes up with first analysis of the data with respect to trends in all these subsections and also gives recommendations as to which certain gas concentrations are supposed to be modeled so that the model results comply with the observations.

- Shouldn't a data set publication primarily focus on presenting the data and, if at all, providing information about the data quality and evaluation / quality assessment activities and results rather than taking already the step of a scientific analysis?

*We take the reviewer's point and have removed the material concerning trends and modelling and retained the remarks which we think will aid readers in their use and interpretation of the timeseries. Subsections 2.1.1 to 2.2.4 have therefore been consolidated into one subsection.*

L261: I note that the data are not freely available per se but require registration and/or a login.

*Unfortunately, this is not likely to change in the near future, but we did have discussed this issue with the editor and they accepted that this would be preferable to not including the data at all.*

L290: Again I am not sure whether the content of this subsection fits with a data set publication paper. If it is for quality assessment then I would have expected a notion into that direction but I could not find one. And therefore this subsection reads like the start of the data analysis.

*The justification for this section is that we want to provide evidence that the data is from the marine sector*

Figure 7:

- It is not clear what the panels a) and b) are in Figure 7. The respective labels a) and b) are missing in the figure.

*Subpanels are now labelled*

- I also suggest to clarify in the caption of the figure the meaning of JFM, MAM and so forth.

*done*

- "DU" is "Dobson Units"? You might want to spell this out in the legend annotation for clarity.

*done*

- What is "Tg" denoting in the four panels of a)? I assume Teragrams?

Done we have removed these text labels

- What is "STT" denoting?

We *have removed these text labels*

L430-436: This paragraph simply reads as an advertisement of the many papers that resulted out of the ACSIS activities - but it has not direct implication to better describing or illustrating the data and their quality themselves.

*We have removed this paragraph*

L440: Which are these "sophisticated techniques"? Please name them and provide a reference as this is important to judge the credibility of the methods used and hence of the data set created.

*We have now cited the relevant paper*

L488: "drift" --> I don't understand what you want to illustrate with this model drift. Is it good to see that the salinity in the upper 1000 meters develops into completely different directions for DFS5.2 compared to for CORE2 (and JRA-55)? I don't understand, whether what you show here is meant to be an indicator of the "quality" or "reliability" of the model

results. Please invest some more writing into this topic if deemed required - also because the next section gets back to "drift".

*Users of the data will need to how their results are influenced by model biases in order to assess their robustness or in order to assess the impact of model biases on the simulated ocean circulation variability. This is now explained in the paragraph.*

Figure 12:

- I am a bit puzzled by a "surface current magnitude" with a unit m s-2; this is an acceleration. Anomalies (as shown) should have the same unit as the regular variable.

- I note that the title underneath the panels on the left says: "Surface speed (m s-1)"; here the unit is correct but the quantity needs to be corrected to "surface current speed anomalies".

*The units for the speed were wrong and have been corrected.*

*The caption for the speed is correct - this is the speed, not the anomaly of the speed (however the colour bar was mislabelled -0.5 to 0.5 which may have confused the reviewer - the colour bar is now labelled correctly 0 to 0.5).*

- What is the rationale to use a classical two-color (blue to red) color bar for the SSH anomalies but a color bar which makes distinction between positive and negative anomalies overly difficult for the surface current speed anomalies. I recommend to change this.

*The variable plotted is the speed, not the anomaly of the speed, so the palette is appropriate. As explained above the reviewer was understandably confused because the range for the speed in the colour bar was wrongly stated as -0.5 to 0.5 instead of 0 to 0.5.*

L566-577: What you write in this paragraph is certainly interesting. But does it support a data set paper or isn't this better to be placed in, e.g., Journal of Geophysical Research - Oceans?

*The first-order interest in these figures is the positions of the Gulf Stream separation and of the NAC, which we believe we have summarised for readers.*

L568/569: The maps shown in the left panels in Figure 12 do not contain any direction information. Therefore this notion about "current shifts direction" is not backed up credibly by the figure as shown.

*As mentioned above, these panels show the absolute surface speed, so it is valid to talk about changes in path.*

L581/582: It is not clear what these T, U, V, W grids are. Is "T" refering to time? Are "U", "V", and "W" refering to eastward positive, northward positive and upward positive (?) components of the 3-D ocean current? Please clarify.

*We have added a more detailed explanation here. The T grid file contains the tracers, including temperature, salinity and surface fields. We confirm that U, V and W are defined to be positive eastwards, northwards and upwards, respectively.*

Figure 13:

This cannot be the sea ice volume - simply because the unit does not fit. CryoSat-2 provides estimates of the sea ice thickness. To compute the sea ice volume one needs to combine the thickness with the sea ice area. Please check what you are showing in this figure and correct it, including the y-axis annotation and the caption. Also in the text (as in the legend) it needs to be "CryoSat-2", i.e. a capital "S".

*Figure 13 shows sea ice volume / grid cell area. This is a common way to present thickness if sea ice were spread homogeneously over the whole grid cell. The unit m and the y-axis annotations are correct. We have multiplied the sea ice thickness with sea ice concentration derived from passive microwave satellite data SSM/I with NASA-Team Bootstrap algorithm (Comiso, 2017). We added this in the manuscript. Thank you for spotting the typo.*

L625 ... Section 5: I recommend to delete this entire section. For me this is blowing up this data set publication beyond the focus it initially had.

I can see that this section contains descriptions of data sets that have been used in the manuscript for, e.g., inter-comparison purposes such as the CryoSat-2 sea ice thickness data set. These additional descriptions and data sets should, however, be put into a subsection, perhaps named "Auxiliary data sets", which should be placed after the introduction. There those data sets that are mandatory to understand your data set publication should be described. Any additional data set descriptions, advertisements, results of preliminary analyses of these that do not inform about data set quality and/or usage limitations should be left out. Alternatively, if you don't like the idea of a separate subsection, then I recommend to include the respective descriptions of the auxiliary data sets in those sections into which these belong; this means for instance that the CryoSat-2 data set description should go into section 4.

*We prefer to retain this section. please see our response to the Reviewer's General Comment 3*

Typos / Editoral Comments:

L50 Typo "may" --> "May"

*Done*

L165: Perhaps add "ARA" after ACSIS for clarity?

*Done*

Figure 3:

- The legend given underneath the figure would benefit from increasing the symbol size and line thickness to better discriminate the different colors.

*Done (the legend also lists the variable in the order (top to bottom) that they are plotted which also aids the reader in discriminating which is which)*

- The y-axis title denotes "chemical species" but with that does not include the wind speed and temperature. I suggest to correct this accordingly.

*Done*

L275: "very wide sector" --> On the web page it says from 110 to 240 degrees direction; you could be more specific in the text.

*Done*

L339: You refer to "Fig 6b" but the panels in Figure 6 do not have labels a) and b).

*These have now been added*

L421: Typo? "bought" --> "brought"

*Done*

L426: Check for blanks ... "below(King, 2023) . On ..."

*Done*

L458: This needs to be "King, 2023", right?

Yes, Changed

L464: I am a bit surprised that the annual low pass filter begins in year 2 and ends in the last but one year. Wouldn't such a filter come into affect already after half a year and also end just half a year before the end of the time series?

*The filter actually has a cut off of 1.8 years so this is why there is a gap of about a year at each end of the timeseries. This is now made clear in the FIgure caption*

L477: "met" --> "meteorological"

*Done*

L478: I think "heat" can be deleted as the fluxes you are refering to are exclusively radiative ones.

*Done*

L492-494: You may consider to remove this last sentence about your expectations.

*Done*

Figure 9: Fonts used are way too small and need to be increased (except the experiment names and variable names, of course).

*Done and new fig added in to new MS*

L521/528: There is no section 4.4.1 in this paper.

*Changed to 3.2.1*

L522: The integration time given is one year shorter than the time for the forcing data set (L514) which extends to 2021 instead of 2020.

This was an error: both integrations finished in 2021.

L528: It might make sense to edit the "1/4 degree" the same way as you write the 1/12 degree - here, elsewhere and in the subsection title.

*We have checked all the occurrences and modifed*

L530: Typo: "how" --> "show"

*Done*

L537: Typo: "from" --> "From"

*Done*

L538: The surface heat flux is ...? Latent and sensible heat fluxes?

*Clarified it is the net heat flux*

Figure 11:

- Again the font size is too small in basically all panels.

*We have increased the font size*

- I suggest to use a), b), ... to denote the panels instead of refering to "bottom left" et cet.

*Done*

 L551: 1/2 needs to be changed to 1/12; I also suggest to write "Time series" instead of "Trends".

*Done*

L567: "surface velocity" --> not consistent to what is shown in the figure = surface current speed anomalies.

*As explained above the variable is surface current  - changed accordingly*

L571: "is missing" --> "is almost missing"

*Done*

L608: "Cyrosat-2" --> "CryoSat-2".

*Done*

What is the source of the data you used?

*The CPOM CryoSat-2 data (Tilling et al., 2018). We mentioned this in original manuscript in Section 5.5: "The method and dataset are detailed in full in Tilling et al., (2018), and monthly sea ice thickness, gridded at 5km, are available from the CPOM data portal http://www.cpom.ucl.ac.uk/csopr/seaice.php." We have altered the text to direct the reader to Section 5.5.*

L767: "ice sheets" --> I don't think I found results of the ice sheets in this data set publication …

*We have removed the reference to ice sheets*

L778: "been described" ? Please check.

*Done*

L834: "2016" --> "(2016)"

*Done*

**Reviewer 2**

The manuscript presents several observational and model datasets for the study of North Atlantic climate variability. The datasets are new and useful and of high quality, and I think they are appropriate for publication in ESSD. The presentation is sometimes unclear. I have some minor comments on the presentation of the datasets and the figures, detailed below.

General comments:

1. The description of the datasets sometimes lack context explaining how they contribute to better understanding the North Atlantic climate and the objectives of ACSIS. This is sometimes specified for the modeling datasets, for example l. 470 "to provide a tool for scientific investigation of the mechanisms of variability of the AMOC and other modes of variability of the Atlantic Ocean". The collection of datasets presented here is very broad, and the manuscript would be easier to follow if the beginning of each dataset subsection contained such reminders (for example 2.1, 2.2 2.3).

*We have added these contextual details at the beginning of subsections as suggested.*

2. The manuscript goes into very different levels of details depending on the subsections, and would benefit from a more consistent style. Several subsections (see below) include detailed analysis that are in my opinion not appropriate for ESSD and could be removed, in order to focus on the description of the data and their limitations.

*We have removed much of the detailed analysis (see responses to specific comments)*

3. The text on several of the figures is too small and sometimes not readable at all.

*We have improved readability of figures where necessary*

4.  Are the atmospheric, ocean and ice modelling datasets meant to be used in synergy? If yes, can you comment on the limitations from using different atmospheric forcing datasets?

*We have added a sentence regarding this in Section 1.1*

SPECIFIC COMMENTS:

L48-49: I think it would be best to write "measurements from 7 aircraft campaigns (N total number of flights)" rather than the range of number of flights per campaign.

*Done*

L155-158: Table 1 mentions aerosol data. Can you also quickly remind here what kind of aerosol data was observed?

*The aerosol data mentioned in Table 1 is listed at the bottom of Table 2 (organic, SO4, NH4, NO3, nss-Cl, possibly the reviewer missed it?)*

L161-162: As a modeler it is often very useful for me to know about the lower detection limit of the instruments, especiallyif these values are reached during the flight. Can this information be added to the table or discussed here, and can you confirm that this information present in the data files?

*We have given values where we can in the column labelled precision 3sigma and have now added a line in the text where Table 2 is first mentioned.*

Figure 1: Please consider putting the Atom flights last in the legend and separate Atom flights more clearly from your ACSIS flights in the legend (grouping/spacing the legend entries and/or adding headers in the legend). -

*Done and new figure added to new MS*

2.1 I found the organization of this section confusing. 2.1 starts with a long text on the flight data without a top subsection, and is followed by very short subsections 2.1.1 to 2.1.3. I think it would be more understandable if it was structured differently. For example: 2.1.1 description of the campaign flights (with Figure 1 and Table 3) 2.1.2 instrumentation 2.1.3 vertical distribution of pollutants, 2.1.4 data archive. I think Table 3 is useful for complementing Figure 1 but I don't see the point of putting it into a separate "bulk analysis" section and I think the associated text has little to do with Table 3, and could instead be attached to the subsection where Figure 2 is found.

*We have followed the reviewer's suggestion and reorganised the subsections along those lines, thank you for the suggestions. With regard to the "bulk analysis" section we have merged that text with the section on vertical distribution of pollutants as suggested. We think Table 3 belongs here because it summarises the data used in the production of Figure 2.*

L205-209: I am not familiar with how these observatories operate so this question might make little sense, but since these observations began before ACSIS can you maybe clarify how this data is new as part of the ACSIS programme, for example how ACSIS contributed to the new data?

*ACSIS funded collection of the data 2016-2021 however previous data are useful to provide context and the entire timeseries is documented and made freely available with this publication*

L216-220: I think this analysis is too detailed for an ESSD paper and could be removed entirely, being replaced at the beginning of 2.2 by a more high-level short sentence on how the CVAO data contributes to ACSIS objectives, see general comments.

*We have removed these lines.*

Figure 3 – The text is very small on the figure

*We have increased the size of the text*

2.2.1 to 2.2.4: These sections go in much more detail than the rest of the subsections, and includes preliminary analysis on trends that could in my opinion be removed (See general comments). You could instead consider showing the observation time series for these species in a new figure, and additional details on the limitations of the data.

*We have merged these sections into one shorter section, concentrating on some general observations which could aid the reader in interpretation of the time series. We also took the opportunity to split Figure 3 into two smaller Figures for clarity.*

2.3 My comments on section 2.2 also aply here (needs a high-level description of the role of this data in ACSIS, maybe clarify the novelty of this data as part of ACSIS, and the analysis might be too detailed for an ESSD paper)

*We have simplified the presentations, removing the Mace Head comparison for ozone and the long-term methane trend fits at Penlee.*

L293 – "NAME" I suppose the model name is missing here? See also next comment.

We've now defined NAME: Numerical Atmospheric-dispersion Modelling Environment from the UK Met Office

2.3.1 The sector analysis is in my opinion too detailed. However, if it is included here; there also needs to be an introduction to why and how the sector analysis was done, and information about the backtrajectory model and its setup in a separate subsection. If I understand correctly, this airmass history modeling dataset was already published, so it could also be removed from this section and included in section 5.

*Users of the data will need to know to what extent the data samples the open ocean sector. We compare the two methods of defining the open ocean (Atlantic) wind sectors here, one using local wind direction, and the other using the NAME model outputs. The approximate agreement between the two methods gives us more confidence in using the local wind direction to define the open ocean sector in subsequent analysis (e.g. CH4 trends).*

Figure 4 – The text is also too small on the figure.

*We have increased the font size.*

L.355-363: Can you remind here at the beginning what this dataset is used for in ACSIS? (See general comment)

*We have moved a later paragraph which explains this to the beginning of the section.*

2.4: Can you also specify here the types of variables predicted by the models and present in the dataset (e.g. trace gases, aerosols, atmospheric dynamics, others)? Right now this description makes it seem like this is only a forced tracer simulation. The description of the dataset in the README file is more clear but I suppose this is only for a subset of the variables: "The following fields are contained in the dataset: O3, NO, NO2, CO, CH4,  4x Stratospheric O3 tracers, and 30x idealised tracers emitted from various locations (15 with a 5-day e-folding lifetime, and 15 with a 30-day e-folding lifetime). Data is provided in mass mixing ratio (kg species/kg air)."

*This is mentioned in subsection 2.4.1 "these include ozone and ozone precursors ($O_3$, NO, $NO_2$, CO and methane) and the idealised tracers used to diagnose transport in the North Atlantic" this is as the Reviewer says, only a subset of the number of chemical species archived. The UKESM model calculates and outputs a large number of dynamical and chemical species and it would be out of the scope of this paper to specify each one of them. For a comprehensive description of the model and its output we will refer the reader to the relevant publication (Archibald et al. 2020, https://doi.org/10.5194/gmd-13-1223-2020) We have therefore replaced:*

*"Atmospheric composition was simulated using the UKCA chemistry module, applying the stratosphere-troposphere chemical mechanism of Archibald et al. (2020) with the 2-moment prognostic aerosol scheme as described in Mulcahy et al. (2020) "*

*With:*
*"UKESM simulations are performed using the StratTrop chemical scheme which simulates the $O_x$, $HO_x$ and $NO_x$ chemical cycles and the oxidation of carbon monoxide, ethane, propane, and isoprene in addition to chlorine and bromine chemistry, including heterogeneous processes on polar stratospheric clouds (PSCs) and liquid sulfate aerosols (SAs). The two-moment GLOMAP-mode aerosol scheme from UKCA (Mulcahy et al.,*

*2020), is used to simulate sulfate and secondary organic aerosol (SOA) formation and is driven*

Figure 7 – I think you can label the panels a-e and specify that a-c are showing seasonal averages in the caption.

*Done*

L469 -  "as a deliverable for WP2.3 of ACSIS" I think this can be removed.

*Done*

Figure 9 and Figure 11 – The text is too small and not readable at all.

*Done - new figures have been prepared and added to the new MS*

Sections 3 and 4 - I am not fully qualified to review these sections but found them well written and clear with no obvious issues.

*Thank you.*

---

## Author Response (AR1)

Many thanks to the Editor and the Reviewers for their efforts with our manuscript. Here are our responses (in highlighted italics)

Editor

Reviewer 1 stated "Section 5: I recommend deleting this entire section."
It seems that whether dispersed throughout the manuscript or centralized in a single section, this information disrupts the flow for readers. I recommend significantly shortening Section 5. Seven pages to describe previously published work is more suited to a review paper than a dataset description. You could condense this into a "Potential Synergy with Previously Published Work" section, where you list studies that can be used in combination with the presented data. This should remain at a high level, for example: "The data presented in Sections X and Y could be used in combination with simulations of aerosol from volcanic eruptions (Arfeuille et al., 2013; Luo, 2016...), also produced under the auspices of ACSIS." In this section, the data you presented in Sections 2 and 3 should be the focus, with auxiliary data mentioned only in terms of how they complement the main datasets. This section should not exceed 1.5-2 pages.

*In response to this point we have taken your advice and cut Section 5 to 1.5 pages and renamed it "Synergies with Previously Published Work". We now only have three subsections, respectively covering volcanic aerosols, coupled simulations, and ice observations, and have made clear the links to sections 2, 3, and 4. We have removed the sections on SST and AMOC observations and corresponding Figures 13 and 14 (of the original MS) as these are now clearly and concisely referenced in the second introductory paragraph of Section 3. We have also removed Table 8 as we realised this information could be condensed and incorporated into the text of Section 5.2. We hope this is now satisfactory.*

*Lines 644-701 of the new MS.*

Reviewer 2 stated, "Section 2.3.1: The sector analysis is, in my opinion, too detailed."
Indeed, it is too detailed for a simple illustration of the dataset. Please focus on one of the two methods to define the open ocean. If you prefer, you can mention that an alternative approach using NAME was also investigated, yielding similar results. Consider removing Figure 4 if its primary purpose is to show the identification of ocean air masses from wind direction. Figure 4b and its accompanying description are not very conclusive and could also be removed. If you'd still like to include a plot of the O3 data, you could add it as a panel in Figure 5, aligned with the methane data. Please also update the plot to include the latest data (currently stopping at early 2021).

*We did remove Figure 4 and condensed the text of section 2.3.1 as suggested. We updated Figure 5 to include the latest data as requested.*

*Lines 295-303 of the new MS.*
*With the removal of Figure 5 and the addition of new Figure 1 as suggested by Reviewer 2 below, the Figure numbers have all changed and we have updated them and references to them in the text as appropriate at many points in the MS.*

Finally, I recommend making all your plotting scripts available in a repository. This would enhance the reproducibility of your article and make it easier for readers to re-use the presented dataset.

*We have created a Zenodo repository (**10.5281/zenodo.13972335**) and supplied a script for each Figure and a README file. This is now referenced in the data availability section.*

*Lines 794-795 of the new MS.*

Reviewer 1

Data supporting the North Atlantic Climate System: Integrated Studies (ACSIS) programme, including atmospheric composition, oceanographic and sea ice observations (2016-2022) and output from ocean, atmosphere, land and sea-ice models (1950-2050)

by A. T. Archibald et al.

Summary: This is a very heterogeneous data set publications wherein the authors try to assemble a suite of physical and chemical observations that in one way or the other have to do with the ACSIS programme. The wealth of the data sets mentioned and described is immense. Overall, I find that the authors solved the challenge to group the different data sets into thematic containers quite well. Naturally, the diversity of the data sets and sources means that the manuscript contains quite a number of tables but this is ok given the heterogeneity of the data sets mentioned. Also, the paper provides a convincing list of links to respective respositories where the data, which are partly a subset of larger collectons, can be accessed. Overall, I find this manuscript useful. I have a number of concerns, though, that I would ask the authors to think about and iterate the manuscript accordingly - as specified in my general comments and also in my specific comments.

General Comments:

GC1: This data set publication contains - to my taste - a too large amount of results from pre-liminary data set analysis that is better to be put into other publications. There are quite some paragraphs in this manuscript as written, which read like an advertizement of the many things that have already been done with the data and/or that could potentially be done with data. I do not find this appropriate for a data set publication of this kind. You find more about this in my specific comments.

*We have toned down much of the preliminary analysis (especially the trend analyses) and the citations of work done in ACSIS as highlighted in the specific comments by both reviewers. Instead, we have tried to confine ourselves to comments which will highlight any limitations of the data or which will aid in their interpretation. Please see responses*

*to specific comments below.*

GC2: At the same time, the manuscript - as written - is overly light when it comes to detail data set quality, reliability and limitations of use. While I understand that the majority of the data sets mentioned here stem from in situ observations, more emphasis on uncertainty sources and potentially limited reliability of the data obtained would immensely assist users in doing a good job when utilizing your data in their research.

*Thank you for the comment, we have done our best to include as much additional information as possible (e.g. Table 2 includes information on precision and uncertainty, we go to some effort to show the origin of the air sampled by the Penlee Point observatory, and model simulations are all ensembles, sampling uncertainty in surface fluxes, model resolution, ice physics etc). We are however ready to respond to any specific lack of clarity or uncertainty that the Reviewer is able to point to.*

GC3: Section 5 contains a - to me somewhat unmotivated - addition of other data sets that may or may not be already published in the context of ACSIS. This I find sub-optimal and I suggest to only keep those data set descriptions of section 5 that immediately have to do with the suite of data that is published with this data set publication AND put the respective information either into a separate section after the introduction or to split section 5 and include the respective information into the specific subsections to which these auxiliary data directly contribute. See my specific comment in this regard

*Please see our response to the Editor's comment above covering the same point. In summary we have cut this down to a very brief description (only 3 short subsections now) of previously published data that is likely to be useful to analyse in conjunction with the data presented in sections 2, 3 and 4.*

*Lines 644-701 of the new MS.*

GC4: Quite a number of the figures and illustrations need further work to make the content readable; a few errors need to be corrected. See my specific and editoral comments in this regard.

*We believe we have addressed these concerns – please see responses to specific comments on the Figures below*

Specific Comments:

L115: Maybe consider to have such a footprint map in this data science paper as well?

*A new schematic map has been added (Figure 1 of the new manuscript)*

L161 / Table 2: There are two Ozone measurement instruments that seem to have been used in succession. How was the inter-instrument calibration carried out to ensure that the ozone measurements of the two instruments are comparable?

*The Thermo Fisher Scientific Model 49i and 2B Technologies Model 205 dual beam photometers operated onboard the FAAM Airborne Laboratory are routinely calibrated against an ozone primary standard (Thermo Fisher Scientific model 49i-PS) maintained by the University of York's COZI laboratory, part of the Wolfson Atmospheric Chemistry Laboratories. This primary standard maintains the traceability of ozone photometers operated by ground-based and airborne facilities of the National Centre for Atmospheric Science to NIST, using the UK National Physical Laboratory's Standard Reference Photometer No 20.*

*The two photometers operated by FAAM have been test-flown alongside each other, sharing a common inlet, to ensure measurement comparability.*

*The figure below illustrates such a test flight demonstrating excellent agreement for the full altitudinal range of the FAAM research aircraft (35000 ft).*

[Figure]

*The following figure illustrates the linear fit of ozone mixing ratios recorded by the two photometers, averaged to 4 seconds (the Model 49i measurement rate) and weighted with uncertainties (3% or 3 ppb), for 28 missions flown by FAAM between 02/07/2019 and 14/02/2020.*

[Figure]

*In some occasions, the ozone comparison between both measurements did not fall within combined uncertainties as shown in the following figure.*

[Figure]

*We have investigated this bias in our laboratories in 2020 and found it is related to the behaviour of the Dewlines/NafionTM water exchange membrane technology employed in the 2B photometer with changing humidity. This issue has also been identified by Eric Hintsa at the NOAA Global Monitoring Laboratory (https://doi.org/10.5194/amt-14-6795-2021), who fly the UCATS payload also utilising 2B*

*Technologies photometers.  Our experience and further discussions with 2B Technologies concluded that the performance water exchange membrane degrade over time under certain conditions.*

*A full description of the 2B Technologies Model 205 ozone photometer can be found in this Zenodo reference [https://doi.org/10.5281/zenodo.7503437](https://doi.org/10.5281/zenodo.7503437) published 04/01/2023.*

*The potential bias is accounted for in our increased ozone measurement uncertainty for the 2B Technologies model 205, namely ±5 ppb or (3%+2 ppb) whichever is greater, and hence* we do not think it is necessary to make any changes to the text.

L190: What is the scientific rationale to perform this grouping? Isn't the density of measurements changing a lot with altitude so that a linear grouping is perhaps not optimal?

*Figure 2 has now been updated so that the heights of the boxes represent the number of observations within each altitude bin (taller box corresponds to more observations). ACSIS flights were planned so that measurements were taken as evenly as possible through the atmosphere (within the range of the aircraft). The 1000 m altitude grouping was chosen as a compromise between showing structural detail in the atmosphere and readability of the plot.*

Figure 2:

- I understand that all six panels show a mixing ratio. However, it would increase the readability of the panels if you'd remove the titles and instead include the respective information into the x-axis caption, i.e. "NO mixing ratio [ppt]" or "CO mixing ratio [ppb]".

*Changed and new figure added to MS.*

- Another area of improvement would be to use the same y-axis scaling. Currently these differ between the top four panels and the bottom two panels.

Changed.

- Also, I count nine bins in the bottom two panels but ten bins in the top four panels. None of these fit to the noted 1000 m bin chosen.
*Measurements of O3, CO, CO2, CH4, were made up to ~9140 m (text has been updated to give correct altitude range now). The NOx instrument cannot give reliable data above 27000 ft/ ~8200 m and so there is no data presented for the top bin range (9000 m – 10000 m).*

- While the bottom two panels seem to fit to the maximum measurement altitude of 7600 m mentioned in the text, the top four panels do not. Please correct the panels and/or correct also the text since it does not fit to these panels.

*This was an error in the text and line 167 been changed to reflect the true maximum altitude observations were made at.*

*Line 169 of the new MS*

- Does the vertical line shown in the boxes denote the mean or the median value?

*Median value. Figure caption has been updated to reflect this.*

*Lines 200-207 of the new MS.*

- Does the horizontal extent of the boxes denote the interquartile range, i.e. from 0.25 to 0.75?

*Yes, and the figure caption has been updated to reflect this.*

*Lines 200-207 of the new MS.*

- Do the bars extending left and right from the box denote 1, 2, or 3 standard deviations?

*The bars extend to the smallest and largest values no more than 1.5 times the interquartile range. The caption has been updated to reflect this.*

*Lines 200-207 of the new MS.*

L193-195: I am not sure I understand correctly. Has the filtering that is mentioned here been done only for the sake of improving the readability of the panels shown? Or were data excluded from the data set? If the latter, what is the scientific rationale to exclude those high mixing ratios?

*The filtering has been done to improve the readability of the panels and the figure caption text has been updated to explain this.*

*Lines 200-207 of the new MS.*

L198-204:

- I am not convinced that the provided ascii files (one kind with comma separation, one

kind with empty space separation) are an overly user-friendly access point to the wealth of data that is going to be published here. Making this data available in netCDF file format would be substantially more useful. The authors might want to motivate why they decided to not make an effort to provide the data in both, ascii text and netCDF file format.

*We have provided the data in both ascii and netcdf formats as described in section 2.1.4: "The merged files are open access and designed to be a tool for an initial exploration of the data and to highlight the breadth of the atmospheric composition data collected during the ACSIS programme. However, for further analysis the original frequency data should be used and details of where these files can be found is included in the header information of the merged files"*

*Lines 214-217 of the new MS.*

- Please check this paragraph for "merge file" vs. "merged file" [correct], and also check for punctuations; one is missing in L203 while one is too much in L204.

*We found a few inconsistencies and have now used "merged" throughout the manuscript.*

*Lines 212-217 of the new MS, heading of column 7 in Table 2 (just after line 190 of the new MS.*

L228+++ (all subsections 2.2.1-2.2.4)

- I am not sure I understand why this paper, dedicated to be a data set publication, comes up with first analysis of the data with respect to trends in all these subsections and also gives recommendations as to which certain gas concentrations are supposed to be modeled so that the model results comply with the observations.

- Shouldn't a data set publication primarily focus on presenting the data and, if at all, providing information about the data quality and evaluation / quality assessment activities and results rather than taking already the step of a scientific analysis?

*We take the reviewer's point and have removed the material concerning trends and modelling and retained the remarks which we think will aid readers in their use and interpretation of the timeseries. Subsections 2.2.1 to 2.2.4 have therefore been consolidated these into one subsection now renamed 2.2.1 in the new MS.*

Lines 235-264 of the new MS.

L261: I note that the data are not freely available per se but require registration and/or a login.

*Unfortunately, this is not likely to change in the near future, but we did have discussed this issue with the editor and they accepted that this would be preferable to not including the data at all.*

*Line 266 of the new MS.*

L290: Again I am not sure whether the content of this subsection fits with a data set publication paper. If it is for quality assessment then I would have expected a notion into that direction but I could not find one. And therefore this subsection reads like the start of the data analysis.

*See our response to the Editor's second comment above. In brief we have substantially pruned this section and removed the Figure 4*

Figure 7:

- It is not clear what the panels a) and b) are in Figure 7. The respective labels a) and b) are missing in the figure.

*Subpanels are now labelled*

*Figure 8, after line 405 of the New MS.*

- I also suggest to clarify in the caption of the figure the meaning of JFM, MAM and so forth.

*Done*

*Line 408-411 of the New MS.*

- "DU" is "Dobson Units"? You might want to spell this out in the legend annotation for clarity.

*Done*

*Line 408-411 of the New MS.*

- What is "Tg" denoting in the four panels of a)? I assume Teragrams?

 *Done we have removed these text labels*

*Figure 8, after line 405 of the New MS.*

- What is "STT" denoting?

We *have removed these text labels*

*Figure 8, after line 405 of the New MS.*

L430-436: This paragraph simply reads as an advertisement of the many papers that resulted out of the ACSIS activities - but it has not direct implication to better describing or illustrating the data and their quality themselves.

*We have removed this paragraph*

*Line 444 of the new MS.*

L440: Which are these "sophisticated techniques"? Please name them and provide a reference as this is important to judge the credibility of the methods used and hence of the data set created.

*We have now cited the relevant paper*

*Lines 447-448 of the new MS.*

L488: "drift" --> I don't understand what you want to illustrate with this model drift. Is it good to see that the salinity in the upper 1000 meters develops into completely different directions for DFS5.2 compared to for CORE2 (and JRA-55)? I don't understand, whether what you show here is meant to be an indicator of the "quality" or "reliability" of the model results. Please invest some more writing into this topic if deemed required - also because the next section gets back to "drift".

*Users of the data will need to how their results are influenced by model biases in order to assess their robustness or in order to assess the impact of model biases on the simulated ocean circulation variability. This is now explained in the paragraph.*

*Lines 505-505 of the new MS.*

Figure 12:

- I am a bit puzzled by a "surface current magnitude" with a unit m s-2; this is an acceleration. Anomalies (as shown) should have the same unit as the regular variable.

- I note that the title underneath the panels on the left says: "Surface speed (m s-1)"; here the unit is correct but the quantity needs to be corrected to "surface current speed anomalies".

*The units for the speed were wrong and have been corrected.*

*The caption for the speed is correct - this is the speed, not the anomaly of the speed (however the colour bar was mislabelled -0.5 to 0.5 which may have confused the reviewer - the colour bar is now labelled correctly 0 to 0.5).*

*Fig. 13 of the new MS – just after line 575.*

- What is the rationale to use a classical two-color (blue to red) color bar for the SSH anomalies but a color bar which makes distinction between positive and negative anomalies overly difficult for the surface current speed anomalies. I recommend to change this.

*The variable plotted is the speed, not the anomaly of the speed, so the palette is appropriate. As explained above the reviewer was understandably confused because the range for the speed in the colour bar was wrongly stated as -0.5 to 0.5 instead of 0 to 0.5.*

*Fig. 13 of the new MS – just after line 575.*

L566-577: What you write in this paragraph is certainly interesting. But does it support a data set paper or isn't this better to be placed in, e.g., Journal of Geophysical Research - Oceans?

*The first-order interest in these figures is the positions of the Gulf Stream separation and of the NAC, which we believe we have summarised for readers.*

*Lines 582-593 of the new MS.*

L568/569: The maps shown in the left panels in Figure 12 do not contain any direction information. Therefore this notion about "current shifts direction" is not backed up credibly by the figure as shown.

*As mentioned above, these panels show the absolute surface speed, so it is valid to talk about changes in path.*

*Fig. 13 of the new MS – just after line 575.*

L581/582: It is not clear what these T, U, V, W grids are. Is "T" refering to time? Are "U", "V", and "W" refering to eastward positive, northward positive and upward positive (?) components of the 3-D ocean current? Please clarify.

*We have added a more detailed explanation here. The T grid file contains the tracers, including temperature, salinity and surface fields. We confirm that U, V and W are defined to be positive eastwards, northwards and upwards, respectively.*

Lines 597-603 of the new MS.

Figure 13:

This cannot be the sea ice volume - simply because the unit does not fit. CryoSat-2 provides estimates of the sea ice thickness. To compute the sea ice volume one needs

to combine the thickness with the sea ice area. Please check what you are showing in this figure and correct it, including the y-axis annotation and the caption. Also in the text (as in the legend) it needs to be "CryoSat-2", i.e. a capital "S".

*Figure 13 shows sea ice volume / grid cell area. This is a common way to present thickness if sea ice were spread homogeneously over the whole grid cell. The unit m and the y-axis annotations are correct. We have multiplied the sea ice thickness with sea ice concentration derived from passive microwave satellite data SSM/I with NASA-Team Bootstrap algorithm (Comiso, 2017). We added this in the manuscript. Thank you for spotting the typo.*

*Lines 640-643 of the new MS.*

L625 ... Section 5: I recommend to delete this entire section. For me this is blowing up this data set publication beyond the focus it initially had.

I can see that this section contains descriptions of data sets that have been used in the manuscript for, e.g., inter-comparison purposes such as the CryoSat-2 sea ice thickness data set. These additional descriptions and data sets should, however, be put into a subsection, perhaps named "Auxiliary data sets", which should be placed after the introduction. There those data sets that are mandatory to understand your data set publication should be described. Any additional data set descriptions, advertisements, results of preliminary analyses of these that do not inform about data set quality and/or usage limitations should be left out. Alternatively, if you don't like the idea of a separate subsection, then I recommend to include the respective descriptions of the auxiliary data sets in those sections into which these belong; this means for instance that the CryoSat-2 data set description should go into section 4.

*Please see our response to the Editor's comment and Reviewer 1's comment GC3 above covering the same point. In summary we have cut this down to a very brief description (only 3 short subsections now) of previously published data that is likely to be useful to analyse in conjunction with the data presented in sections 2, 3 and 4.*

Typos / Editoral Comments:

L50 Typo "may" --> "May"

*Done*

*Line 50 of the new MS.*

L165: Perhaps add "ARA" after ACSIS for clarity?

*Done*

*Line 180 of the new MS.*

Figure 3:

- The legend given underneath the figure would benefit from increasing the symbol size and line thickness to better discriminate the different colors.

*Done (the legend also lists the variable in the order (top to bottom) that they are plotted which also aids the reader in discriminating which is which)*

Figure 4, just after line 198, and lines 200-207, of the new MS

- The y-axis title denotes "chemical species" but with that does not include the wind speed and temperature. I suggest to correct this accordingly.

*Done*

*Fig 4 just after line 255, and lines 257-259 of the new MS.*

L275: "very wide sector" --> On the web page it says from 110 to 240 degrees direction; you could be more specific in the text.

*Done*

*Line 280 of the new MS.*

L339: You refer to "Fig 6b" but the panels in Figure 6 do not have labels a) and b).

*These have now been added*

*Figure 7 of the new MS, lines 332-349.*

L421: Typo? "bought" --> "brought"

*This paragraph has been removed in response to the Reviewer's comments*

*Just after line 432 of the new MS.*

L426: Check for blanks ... "below(King, 2023) . On ..."

*Done*

*Line 438 of the new MS.*

L458: This needs to be "King, 2023", right?

Yes, Changed

Line 467 of the new MS.

L464: I am a bit surprised that the annual low pass filter begins in year 2 and ends in the last but one year. Wouldn't such a filter come into affect already after half a year and also end just half a year before the end of the time series?

*The filter actually has a cut off of 1.8 years so this is why there is a gap of about a year at each end of the timeseries. This is now made clear in the FIgure caption*

*Lines 472-474 fo the new MS.*

L477: "met" --> "meteorological"

*Done*

*Line 479 of the new MS.*

L478: I think "heat" can be deleted as the fluxes you are refering to are exclusively radiative ones.

*Done*

*Line 494 of the new MS.*

L492-494: You may consider to remove this last sentence about your expectations.

*Done*

*After line512 of the new MS.*

Figure 9: Fonts used are way too small and need to be increased (except the experiment names and variable names, of course).

*Done and new fig added in to new MS*

*New Figure 10 after line 513 of the new MS.*

L521/528: There is no section 4.4.1 in this paper.

*Changed to 3.2.1*

*Line 538 of the new MS.*

L522: The integration time given is one year shorter than the time for the forcing data set (L514) which extends to 2021 instead of 2020.

*This was an error: both integrations finished in 2021.*

*Line 539 of the new MS.*

L528: It might make sense to edit the "1/4 degree" the same way as you write the 1/12 degree - here, elsewhere and in the subsection title.

*We have found 17 occurrences and modified accordingly*

L530: Typo: "how" --> "show"

*Done*

*Line 547 of the new MS.*

L537: Typo: "from" --> "From"

*Done*

*Line 554 of the new MS.*

L538: The surface heat flux is ...? Latent and sensible heat fluxes?

*Clarified it is the net heat flux*

*Line 555 of the new MS.*

Figure 11:

- Again the font size is too small in basically all panels.

*We have increased the font size*

*Figure 12 of the new MS, after line 566*

- I suggest to use a), b), ... to denote the panels instead of refering to "bottom left" et cet.

*Done*

*Figure 12 of the new MS, after line 566*

 L551: 1/2 needs to be changed to 1/12; I also suggest to write "Time series" instead of "Trends".

*Done*

*Line 568 of the new MS.*

L567: "surface velocity" --> not consistent to what is shown in the figure = surface current speed anomalies.

*As explained above the variable is surface current  - changed accordingly*

*Line 584 of the new MS*

L571: "is missing" --> "is almost missing"

*Done. We now say it is "extremely faint"*

*Line 588 of the new MS.*

L608: "Cyrosat-2" --> "CryoSat-2".

*Done*

*Line 627 of the new MS.*

What is the source of the data you used?

*The CPOM CryoSat-2 data (Tilling et al., 2018). We mentioned this in original manuscript in Section 5.5: "The method and dataset are detailed in full in Tilling et al., (2018), and monthly sea ice thickness, gridded at 5km, are available from the CPOM data portal http://www.cpom.ucl.ac.uk/csopr/seaice.php." This is now in Section 5.3 of the new MS*

*We have altered the text to direct the reader to Section 5.3 of the new MS.*

*We also provide the source of the data in the text Lines 627-628 of the new MS, and the cation of Figure 14, lines 641-644 of the new MS.*

L767: "ice sheets" --> I don't think I found results of the ice sheets in this data set publication …

*We have explained that ice sheets were covered in ACSIS but not in the present MS.*

*Line 708 of the new MS.*

L778: "been described" ? Please check.

*Done. We have rewriten "is described"*

*Lines 719-720 fo the new MS.*

L834: "2016" --> "(2016)"

*Done*

*Line 775 of the new MS.*

**Reviewer 2**

The manuscript presents several observational and model datasets for the study of North Atlantic climate variability. The datasets are new and useful and of high quality, and I think they are appropriate for publication in ESSD. The presentation is sometimes unclear. I have some minor comments on the presentation of the datasets and the figures, detailed below.

General comments:

1. The description of the datasets sometimes lack context explaining how they contribute to better understanding the North Atlantic climate and the objectives of ACSIS. This is sometimes specified for the modeling datasets, for example l. 470 "to provide a tool for scientific investigation of the mechanisms of variability of the AMOC and other modes of variability of the Atlantic Ocean". The collection of datasets presented here is very broad, and the manuscript would be easier to follow if the beginning of each dataset subsection contained such reminders (for example 2.1, 2.2 2.3).

*We have added these contextual details at the beginning of subsections as suggested.*

*Section 2.1: Lines 152-154 of the new MS*
*Section 2.2: lines 222-224 of the new MS*
*Section 2.3: line 276 of the new MS*
*Section 2.4: 364-370 of the new MS*
*Section 3.1: lines 446-451 of the new MS*
*Section 3.2: lines 477-482 of the new MS*
*Section 4.1: lines 613-614 of the new MS*

2. The manuscript goes into very different levels of details depending on the subsections, and would benefit from a more consistent style. Several subsections (see below) include detailed analysis that are in my opinion not appropriate for ESSD and could be removed, in order to focus on the description of the data and their limitations.

*We have removed much of the detailed analysis (see responses to Editor's comments and Reviewer 1 specific comments above and Reviewer 2 specific comments below).*

3. The text on several of the figures is too small and sometimes not readable at all.

*We have improved readability of figures where necessary (see responses to specific comments for Reviewer1 above and Reviewer 2 below)*

4. Are the atmospheric, ocean and ice modelling datasets meant to be used in synergy? If yes, can you comment on the limitations from using different atmospheric forcing datasets?

*We have added a sentence regarding this in Section 1.1*

*Lines 123-127 of the new MS.*

SPECIFIC COMMENTS:

L48-49: I think it would be best to write "measurements from 7 aircraft campaigns (N total number of flights)" rather than the range of number of flights per campaign.

*Done*

*Line 48 of the new MS.*

L155-158: Table 1 mentions aerosol data. Can you also quickly remind here what kind of aerosol data was observed?

*The aerosol data mentioned in Table 1 is listed at the bottom of Table 2 (organic, SO4, NH4, NO3, nss-Cl), possibly the reviewer missed it?*

L161-162: As a modeler it is often very useful for me to know about the lower detection limit of the instruments, especiallyif these values are reached during the flight. Can this information be added to the table or discussed here, and can you confirm that this information present in the data files?

*We have given values where we can in the column labelled precision 3sigma and have now added a line in the text where Table 2 is first mentioned.*

*Lines 187-188 of the new MS.*

Figure 1: Please consider putting the Atom flights last in the legend and separate Atom flights more clearly from your ACSIS flights in the legend (grouping/spacing the legend entries and/or adding headers in the legend). -

*Done and new Figure 1added to new MS after line 128 (subsequent Figure numbers have been increased in consequence – references to Figures in the text have been updated accordingly)*

2.1 I found the organization of this section confusing. 2.1 starts with a long text on the flight data without a top subsection, and is followed by very short subsections 2.1.1 to 2.1.3. I think it would be more understandable if it was structured differently. For example: 2.1.1 description of the campaign flights (with Figure 1 and Table 3) 2.1.2 instrumentation 2.1.3 vertical distribution of pollutants, 2.1.4 data archive. I think Table 3 is useful for complementing Figure 1 but I don't see the point of putting it into a separate "bulk analysis" section and I think the associated text has little to do with Table 3, and could instead be attached to the subsection where Figure 2 is found.

*We have followed the reviewer's suggestion and reorganised the subsections along those lines, thank you for the suggestions. With regard to the "bulk analysis" section we have merged that text with the section on vertical distribution of pollutants (Section 2.1.3) as suggested. We think Table 3 belongs in Section 2.1.3 because it summarises the data used in the production of Figure 2.*

*Lines 151-219 of the new MS*

L205-209: I am not familiar with how these observatories operate so this question might make little sense, but since these observations began before ACSIS can you maybe

clarify how this data is new as part of the ACSIS programme, for example how ACSIS contributed to the new data?

*ACSIS funded collection of the data 2016-2021 however previous data are useful to provide context and the entire timeseries is documented and made freely available with this publication*

L216-220: I think this analysis is too detailed for an ESSD paper and could be removed entirely, being replaced at the beginning of 2.2 by a more high-level short sentence on how the CVAO data contributes to ACSIS objectives, see general comments.

*We have removed these lines.*

*Just after line 234 of the new MS.*

Figure 3 – The text is very small on the figure

*We have increased the size of the text*

*Figs 4 and 5 of the new MS after lines 255 and 259 respectively.*

2.2.1 to 2.2.4: These sections go in much more detail than the rest of the subsections, and includes preliminary analysis on trends that could in my opinion be removed (See general comments). You could instead consider showing the observation time series for these species in a new figure, and additional details on the limitations of the data.

*We have merged these sections into one shorter section, concentrating on some general observations which could aid the reader in interpretation of the time series. We also took the opportunity to split Figure 3 into two smaller Figures for clarity.*

*Lines 235-264 of the new MS.*

2.3 My comments on section 2.2 also aply here (needs a high-level description of the role of this data in ACSIS, maybe clarify the novelty of this data as part of ACSIS, and the analysis might be too detailed for an ESSD paper)

*We have simplified the presentations, removing the Mace Head comparison for ozone and the long-term methane trend fits at Penlee.*

*Lines 296-319 of the new MS.*

L293 – "NAME" I suppose the model name is missing here? See also next comment.

*We've now defined NAME: Numerical Atmospheric-dispersion Modelling Environment from the UK Met Office*

*Line 299 of the new MS*

2.3.1 The sector analysis is in my opinion too detailed. However, if it is included here; there also needs to be an introduction to why and how the sector analysis was done, and information about the backtrajectory model and its setup in a separate subsection. If I understand correctly, this airmass history modeling dataset was already published, so it could also be removed from this section and included in section 5.

*On the advice of the Editor (see response to second Editor comment above) we have drastically cut down the detail of this section, including removing Figure 4.*

Figure 4 – The text is also too small on the figure.

*Figure 4 has been removed.*

*just after line 304 of the new MS.*

L.355-363: Can you remind here at the beginning what this dataset is used for in ACSIS? (See general comment)

*We have added some text covering this.*

*Lines 364-370 of the new MS.*

2.4: Can you also specify here the types of variables predicted by the models and present in the dataset (e.g. trace gases, aerosols, atmospheric dynamics, others)? Right now this description makes it seem like this is only a forced tracer simulation. The description of the dataset in the README file is more clear but I suppose this is only for a subset of the variables: "The following fields are contained in the dataset: O3, NO, NO2, CO, CH4,  4x Stratospheric O3 tracers, and 30x idealised tracers emitted from various locations (15 with a 5-day e-folding lifetime, and 15 with a 30-day e-folding lifetime). Data is provided in mass mixing ratio (kg species/kg air)."

*This is mentioned in subsection 2.4.1 "these include ozone and ozone precursors ($O_3$, NO, $NO_2$, CO and methane) and the idealised tracers used to diagnose transport in the North Atlantic" this is as the Reviewer says, only a subset of the number of chemical species archived. The UKESM model calculates and outputs a large number of dynamical and chemical species and it would be out of the scope of this paper to specify each one of them. For a comprehensive description of the model and its output we will refer the reader to the relevant publication (Archibald et al. 2020, [https://doi.org/10.5194/gmd-13-1223-2020](https://doi.org/10.5194/gmd-13-1223-2020)) We have therefore replaced:*

*"Atmospheric composition was simulated using the UKCA chemistry module, applying the stratosphere-troposphere chemical mechanism of Archibald et al. (2020) with the 2-moment prognostic aerosol scheme as described in Mulcahy et al. (2020)"*

*With:*
*"UKESM simulations were performed using the StratTrop chemical scheme which simulates the $O_x$, $HO_x$ and $NO_x$ chemical cycles and the oxidation of carbon monoxide, ethane, propane, and isoprene in addition to chlorine and bromine chemistry, including*

*heterogeneous processes on polar stratospheric clouds (PSCs) and liquid sulfate aerosols (SAs). The two-moment GLOMAP-mode aerosol scheme from UKCA (Mulcahy et al., 2020), is used to simulate sulfate and secondary organic aerosol (SOA) formation and is driven by prescribed oxidant fields. For further details on UKESM chemistry and aerosols scheme the reader is referred to Archibald et al. (2020)."*

*Lines 376-381 of the new MS.*

Figure 7 – I think you can label the panels a-e and specify that a-c are showing seasonal averages in the caption.

*Done*

*Figure 8 of the new MS, just after line 405 and Figure caption lines 408-411.*

L469 - "as a deliverable for WP2.3 of ACSIS" I think this can be removed.

*Done*

*Line 486 of the new MS.*

Figure 9 and Figure 11 – The text is too small and not readable at all.

*Done - new figures 10 (after line 513) and 12 (after line 566) have been prepared and added to the new MS.*

Sections 3 and 4 - I am not fully qualified to review these sections but found them well written and clear with no obvious issues.

*Thank you.*

---

## Author Response (AR2)

Dear Editor, Please find our responses to your comments are made in italic text below.

**Public justification (visible to the public if the article is accepted and published)**:
Dear Authors,

Thank you for this revision. I am pleased to accept the manuscript for publication, pending the minor revisions listed below.

Section 2.3.1 still requires additional justification:

- Please begin with a sentence clarifying what you aim to identify through NAME or local wind direction (e.g., Atlantic air masses) and why this is important.

*At lines 275-6 of the new MS we added:*
*'We are particularly interested in the North Atlantic air mass at this coastal location, as this represents the background condition for the UK during the typical southwesterly conditions.'*

- Please rephrase the final sentence to provide a more conclusive statement, such as: "We conclude that the Atlantic air mass (?) can be identified from the local wind direction when ranging from X to Y, and we will use this definition going forward..."

*At lines 296-7 of the new MS we added:*
*" We conclude that the North Atlantic air mass can reasonably be identified from the local wind direction between 210° and 260°, and we use this definition in section 2.3.2 below."*

- In Section 2.3.2, you briefly state that the all-direction methane concentration is higher than that in the southwest sector. Could you elaborate on the meaning of this finding and its implications? Without this, the analysis may seem insufficiently justified.

*At lines 304-306 we added:*
*'We expect measurements from the southwest wind sector to be more representative of the Atlantic and so background Northern Hemisphere. That the all-direction mean mixing ratio is higher reflects local and regional emissions of methane."*

For improved readability, please merge Figures 4 and 5 into a full-page figure and update it with the latest data.

*Done.*

Additionally, for these figures:

- Ensure no overlap between the y-axis tick labels.

*Done.*

- Remove the redundant y labels (e.g., "Variable" in Figure 4) as they add little value. *Done.*

- Remove the box separating the y-axis from the plotting area.

*Done.*

- Replace "TEMP" with "T" or "TA" (for air temperature) since "TEMP" is neither an acronym nor a chemical formula.

*Done.*

- In the caption, please add the short name of each variable (e.g., "From top: wind speed (WS), air temperature (TA), ozone (O3)...").

*Done.*

- Consider adding the variable name and unit as either the y-axis label (stacked if necessary) or as a panel label within each plot, instead of using a legend. To separate name and unit, use "WS [m s-1]" or "WS (m s-1)" rather than "WS_m s-1."

*Done.*

For Figure 7 and all other multi-panel figures, please follow the recommendation for panel labels: https://www.earth-system-science-data.net/submission.html#figurestables "Labels of panels must be included with brackets around letters in lowercase (e.g., (a), (b), etc.)."

*Done (Figs 6, 7 and 8 of the new MS are updated versions).*

Line 606: A closing parenthesis is missing at the end of the line.

*Done*

To avoid further delays, I will not need to review the manuscript after this point.

*Thank you.*

Please conduct a final proofread to address any remaining issues, harmonize the figures, and ensure all captions include essential information for readers. Additionally, verify that all links are functional.

*Done.*

Thank you for your efforts and patience.

Kind regards,
Baptiste Vandecrux